# Tackling GNARLy Problems: Graph Neural Algorithmic Reasoning Reimagined through Reinforcement Learning

## Abstract

Neural algorithmic reasoning (NAR) is a paradigm that trains neural networks to execute classic algorithms by supervised learning. Despite its successes, important limitations remain: inability to construct valid solutions without post-processing and to reason about multiple correct ones, poor performance on combinatorial NP-hard problems, and inapplicability to problems for which strong algorithms are not yet known. To address these limitations, we reframe the problem of learning algorithm trajectories as a Markov decision process, which imposes structure on the solution construction procedure and unlocks the powerful tools of imitation and reinforcement learning (RL). We propose the GNARL framework, encompassing the methodology to translate problem formulations from NAR to RL and a learning architecture suitable for a wide range of graph-based problems. We achieve very high graph accuracy results on several CLRS-30 problems and performance matching or exceeding much narrower NAR approaches for NP-hard problems. Remarkably, GNARL remains applicable when no expert algorithm is available, though reward-driven learning can exhibit greater variability than direct supervision.

## 1 Introduction

Neural Algorithmic Reasoning (NAR) is a framework that aims to emulate the steps of an algorithm using neural networks trained on the labelled frame-by-frame states of the computational process. It has been used primarily for imitating algorithms of polynomial complexity, such as classic algorithms for sorting and searching, with the CLRS-30 Benchmark (Veličković et al., 2022) driving progress in recent years. The sequential nature of NAR enables better reasoning and generalisation capabilities, compared to "one-shot" prediction of an algorithm's outputs based only on inputs (Veličković & Blundell, 2021).

However, several critical limitations of the standard NAR blueprint exist. Firstly, typical NAR pipelines struggle with ensuring globally valid solutions without significant post-processing, and cannot reason about multiple equivalent solutions (Kujawa et al., 2025). Secondly, attempts to apply NAR to NP-hard problems have so far relied on highly specialised approaches (Georgiev et al., 2024a; He & Vitercik, 2025), demanding significant engineering work and sacrificing generality in the process. Lastly, the supervised NAR approach is inherently limited to the expert algorithms it is trained on, and thus cannot be applied to discover algorithms for new problems where an expert does not exist, or improve on the performance of imperfect experts.

The key insight of our work is that the NAR blueprint, which breaks down an algorithmic trajectory into a series of steps (called "hints") with a defined feature space, can be viewed instead as a trajectory in a Markov Decision Process (MDP), the mathematical formalism behind Reinforcement Learning (RL). In standard NAR, trajectories are learned by supervised learning of internal algorithm states; we change this to an MDP formulation to unlock the use of powerful RL tools for that same NAR problem. The correspondence, summarised in Figure 1A, allows us to address several key limitations of NAR: (i) MDP formulations provide a skeleton for ensuring valid solutions by construction and acceptance of several equivalent solutions; (ii) we unify polynomial and NP-hard graph problems under a framework that, unlike existing approaches, retains good performance on the latter class while being general; (iii) we enable NAR to go beyond known algorithms by implicit learning of new ones using only a reward signal.

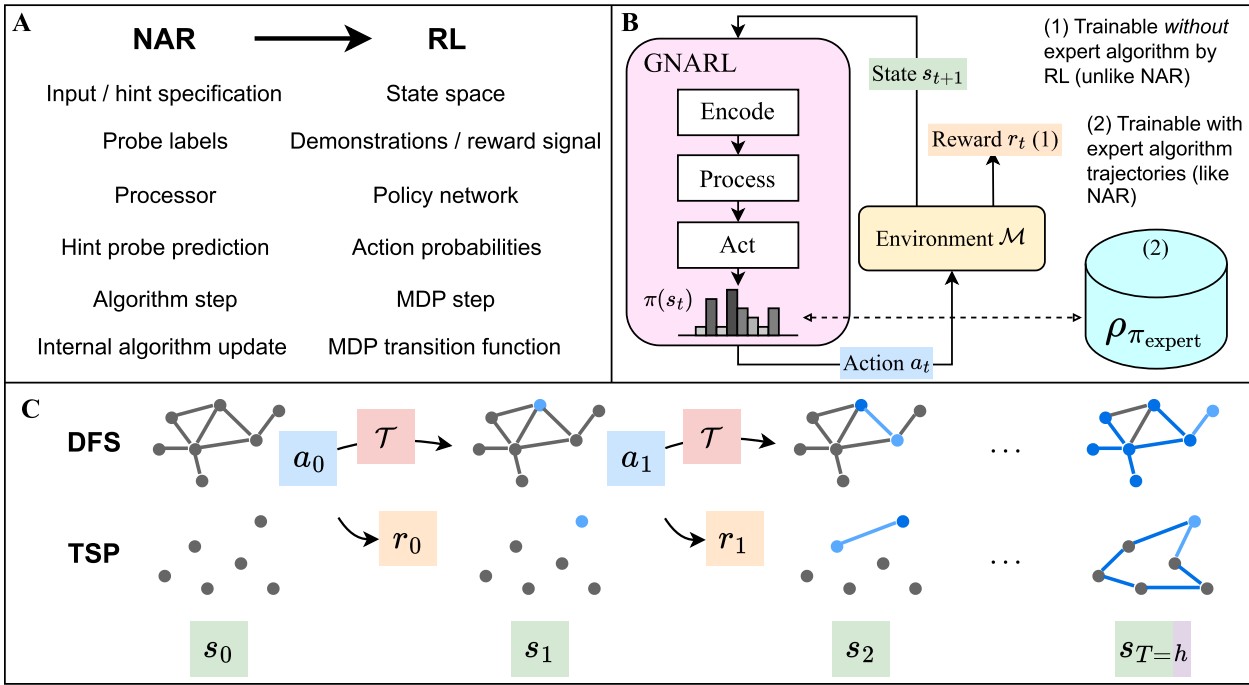

Figure 1: **A.** Key correspondences leveraged by GNARL to cast NAR as an RL problem. **B.** Unlike standard NAR, GNARL is trainable without an expert algorithm by using a reward signal. **C.** Examples of the MDP $\mathcal{M} = \langle\, S\,,\, \mathcal{A}\,,\, \mathcal{T}\,,\, R\,,\, h\,\rangle$ for a polytime solvable and NP-hard problem. At each step, a node is selected, the transition function yields the next state, and a reward is obtained.

Our contributions are as follows: (1) We propose the Graph Neural Algorithmic Reasoning with Reinforcement Learning (**GNARL**) framework that relies on the insights listed above to reframe NAR as an RL problem. To the best of our knowledge, our work is the first to model NAR as an MDP in order to apply RL to it. (2) We build a general learning architecture that is broadly applicable for graph problems in both P and NP. (3) We carry out an extensive evaluation demonstrating that GNARL can construct valid solutions without post-processing, achieves comparative or better performance on NP-hard problems relative to existing problem-specific NAR methods, and is applicable to new problems even in the absence of an expert algorithm.

## 2 Related Work

### 2.1 Neural Algorithmic Reasoning

Neural Algorithmic Reasoning (NAR) is a field introduced by Veličković & Blundell (2021) that targets learning to execute algorithms using neural networks. Unlike approaches that learn direct input-output mappings, NAR models are trained on intermediate steps (*trajectories*) of algorithms, ultimately achieving stronger reasoning and out-of-distribution (OOD) generalisation (Veličković et al., 2020; Mahdavi et al., 2023). NAR has advantages over traditional algorithms when handling real-world data, as it can integrate priors to process high-dimensional, noisy, and unstructured data, generalising beyond low-dimensional inputs (Deac et al., 2021; Panagiotaki et al., 2025).

To date, most applications of NAR have been on algorithms of polynomial complexity (P), predominantly those in the CLRS-30 Benchmark (Veličković et al., 2022; Ibarz et al., 2022). The supervised approach in early NAR works (Veličković et al., 2022; 2020) is limited by the requirement for ground-truth labels, inherently restricting training for NP-hard problems to small examples or algorithmic approximation. Recent methods using NAR with self-supervised learning (Bevilacqua et al., 2023; Rodionov & Prokhorenkova, 2023), transfer learning (Xhonneux et al., 2021), fixed point methods (Xhonneux et al., 2024; Georgiev et al., 2024b), and

reinforcement learning (Deac et al., 2021) focus on problems in P. Attempts to apply NAR to NP-hard tasks have been restricted to specific problems (He & Vitercik, 2025), or are outperformed by simple heuristics (Georgiev et al., 2024a). Furthermore, these approaches do not guarantee that the solutions produced are valid under the constraints of the problem, requiring augmentations such as extensive beam search at runtime to ensure valid outputs.

For many problems, there are multiple correct solutions, e.g., as in depth-first search (DFS). However, the CLRS-30 Benchmark relies on a single solution induced by a pre-defined node order. The typical metric reported for NAR models is the *node accuracy*, or micro-$F_1$ score, which corresponds to the mean accuracy of predicting the label of each node independently (Veličković et al., 2022). Even though recent NAR models achieve high node accuracy (Veličković et al., 2022; Bevilacqua et al., 2023; Bohde et al., 2024), this metric ignores the fact that a single incorrect prediction can completely corrupt a solution: for a shortest path problem, one incorrect predecessor label could render all paths infinite. Alternatively, the graph accuracy metric measures the percentage of *graphs* for which *all* labels are correctly predicted (Minder et al., 2023). Notably, NAR models score extremely poorly in this more realistic metric, with Kujawa et al. (2025) struggling to achieve beyond 50% for OOD graph sizes.

## 2.2 RL for Graph Combinatorial Optimisation

Machine learning approaches to combinatorial optimisation have gained popularity in recent years. The goal is to replace heuristic hand-designed components with learned knowledge in the hope of obtaining more principled and optimal algorithms (Bengio et al., 2021; Cappart et al., 2023; Berto et al., 2025). This area is also referred to as Neural Combinatorial Optimisation (NCO). Particularly relevant to this paper are works that use RL to automatically discover, by trial-and-error, heuristic solvers that generate approximate solutions (Darvariu et al., 2024).

The problems treated in NCO are NP-hard or, even when lacking a formal complexity characterisation, clearly computationally intractable. The approach has mainly been applied to canonical problems such as the Travelling Salesperson Problem (Kwon et al., 2020), Maximum Cut (Khalil et al., 2017), and Maximum Independent Set (Ahn et al., 2020). With few exceptions, the performance of RL-discovered solvers still lags behind traditional ones. The case for RL becomes stronger for problems that currently lack powerful solvers, such as Robust Graph Construction (RGC) (Darvariu et al., 2021a) or Molecular Discovery (You et al., 2018), for which an RL-discovered solver is able to achieve high-quality results compared to simple heuristics. Furthermore, Yehuda et al. (2020) highlight that polytime samplers cannot solve NP-hard problems optimally when trained using supervised learning, allowing only approximate solutions. This analysis does not apply to RL, further motivating its use for NP-hard problems.

## 2.3 NAR-NCO Relationship and Novelty of GNARL

Despite the many shared goals of the NAR and NCO communities (Cappart et al., 2023), their literatures to date have mostly been disjoint, without substantial overlaps. The contributions of our paper should also be understood as bridging the knowledge gap between the two communities.

Relative to NAR methods, GNARL introduces the structured MDP approach and RL/IL learning mechanisms. To the best of our knowledge, and as is reflected by the NAR works surveyed above, this approach is entirely novel in the NAR literature. As we demonstrate, GNARL achieves significant improvements over standard NAR methods in performance and generality. By addressing the key limitations of NAR, we render it applicable for combinatorial optimisation problems, with which the framework has historically struggled.

In the NCO literature, MDP formulations and policies trained with RL are relatively commonplace, and we do not claim novelty from a learning architecture standpoint. Here, our work highlights the value of learning from expert algorithm demonstrations. While this is the de facto approach in NAR, it remains relatively uncommon in NCO compared to purely reward-driven RL-based approaches (Berto et al., 2025). Furthermore, in the spirit of NAR, GNARL is designed to be a general-purpose framework with straightforward application to new problems, and not intended to compete with problem-specific NCO pipelines. This makes significant

progress towards the challenge indicated by Cappart et al. (2023), who argue that integrating graph neural networks for combinatorial optimisation requires a framework that abstracts technical details.

## 3 Background

### 3.1 Neural Algorithmic Reasoning

In NAR (Veličković et al., 2020; 2022), a ground-truth algorithm $\mathtt{A}$ (e.g. DFS, Bellman-Ford) generates a sequence of intermediate steps $\{\mathbf{y}^{(t)} = \mathtt{A}(G_t)\}_{t=0}^{T-1}$, with final output $\mathbf{y}^{(T)} = \mathtt{A}(G_T)$. Here $G_t = (V_t, E_t)$ corresponds to the input graph at each step $t$, with $V_t$ and $E_t$ denoting the sets of nodes and edges, and $\mathbf{x}_v^{(t)}$, $v \in V_t$ and $\mathbf{e}_{uv}^{(t)}$, $(u, v) \in E_t$, their associated features. For $t > 0$, each $G_t$ is generated by the algorithmic execution at the previous step $t - 1$, and $G_0$ corresponds to the initial input to the algorithm. Instead of training on the final output to learn a direct input-output mapping $f : G_0 \to \mathtt{A}(G_T)$, NAR also uses the intermediate steps (i.e., *hints*) as supervision signals to learn the entire algorithmic *trajectory*. An important and non-trivial aspect of this framework is finding the right hints, which should contain enough information to guide the model towards correctly approximating the algorithmic trajectory, while avoiding unnecessary complexity.

The standard NAR architecture iteratively applies Graph Neural Networks (GNNs) with intermediate supervision signals following the 'encode-process-decode' paradigm (Hamrick et al., 2018). In this paradigm, input features are first encoded by the network, $\mathbf{z}_v^{(t)} = \mathrm{enc}_V(\mathbf{x}_v^{(t)})$ and $\mathbf{z}_{uv}^{(t)} = \mathrm{enc}_E(\mathbf{e}_{uv}^{(t)})$, then passed through a Message-Passing Neural Network (MPNN) (Gilmer et al., 2017) that iteratively updates latent features $\mathbf{h}^{(t)}$, then are finally decoded into outputs. At $t = 0$, $\mathbf{h}^{(0)}$ are initialised, and then each subsequent iteration updates $\mathbf{h}^{(t)}$ following:

$$\mathbf{m}_v^{(t)} = \sum_{u \in \mathcal{N}(v)} M_t[\mathbf{h}_v^{(t-1)} \parallel \mathbf{z}_v^{(t)}, \ \mathbf{h}_u^{(t-1)} \parallel \mathbf{z}_u^{(t)}, \ \mathbf{z}_{uv}^{(t)}],$$
$$\mathbf{h}_v^{(t)} = U_t(\mathbf{h}_v^{(t-1)} \parallel \mathbf{z}_v^{(t)}, \ \mathbf{m}_v^{(t)}), \tag{1}$$

where each node $v$ receives messages from its neighbours $\mathcal{N}(v)$, $M_t$ is a learnable message function, and $U_t$ is a learnable update function. A decoder maps the embeddings $\mathbf{h}_v^{(t)}$ to outputs $\hat{\mathbf{y}}^{(t)}$. At each iteration, the decoded output $\hat{\mathbf{y}}^{(t)}$ becomes the input for the next step at $t + 1$, until the sequence of $T$ iterations has been completed. NAR relies on step-wise supervision by aligning $\hat{\mathbf{y}}^{(t)}$ with the internal states of the algorithm, and final-task supervision by aligning $\hat{\mathbf{y}}^T$ with the final output $\mathbf{y}^T$. The number of steps $T$ is typically fixed and given by the algorithm. In NAR, these trajectories are supervised with expert labels and do not use an MDP.

The explicit trajectory alignment in the NAR training process fundamentally aims to improve out-of-distribution size generalisation in neural networks. NAR-based models are typically evaluated on OOD data points, where models trained on small instances are tested on significantly larger graph sizes. By learning algorithmic priors as inductive biases, network predictions are restricted to algorithmically meaningful trajectories rather than random data correlations. As a result, NAR models can generalise robustly and maintain accuracy beyond the training distribution.

### 3.2 Markov Decision Processes and Solution Methods

A Markov Decision Process is a tuple $\mathcal{M} = \langle S, \mathcal{A}, \mathcal{T}, R, h \rangle$, where i) $S$ is a set of states; ii) $\mathcal{A}$ is the set of all actions, with $\mathcal{A}(s) \subseteq \mathcal{A}$ being the set of available actions in state $s$; iii) $\mathcal{T} : S \times \mathcal{A} \times S \to [0, 1]$ is the transition probability function; iv) $R : S \times \mathcal{A} \to \mathbb{R}$ is the reward function; and v) $h$ is the horizon. A solution to an MDP is a policy $\pi$ mapping each state $s \in S$ to a probability distribution over actions, i.e. $\pi : S \to \Delta(\mathcal{A})$ where $\Delta(\mathcal{A})$ denotes the probability simplex over $\mathcal{A}$. The optimal policy $\pi^*$ maximises the expected cumulative reward, i.e., $\pi^* = \arg\max_\pi \mathbb{E}_\pi \left[ \sum_{t=1}^h R(s_t, a_t) \mid a_t \sim \pi(\cdot \mid s_t) \right]$.

Reinforcement learning is an approach for solving MDPs using trial-and-error interactions with an environment to learn an optimal policy. The environment is modelled by the MDP $\mathcal{M}$, and represents the world in which the agent acts, producing successor states and rewards when the agent performs an action. RL algorithms

aim to improve a policy $\pi$ by taking actions in the environment and using the reward as a feedback signal to update the policy. Typical reinforcement learning algorithms include Q-learning, policy gradient methods, and actor-critic methods (Sutton & Barto, 2018). Actor-critic methods simultaneously learn an actor policy $\pi : S \to \Delta(\mathcal{A})$, and a critic function $\delta : S \to \mathbb{R}$, which estimates the value of a given state. Proximal Policy Optimisation (PPO) (Schulman et al., 2017) is a popular method, using a clipped surrogate objective which updates the actor policy while ensuring that the updated policy does not deviate too much from the previous one. At each iteration, PPO samples experience tuples $\langle s_t, a_t, r_t, s_{t+1} \rangle$ using the current policy $\pi_i$, and the advantage estimator $\hat{A}(s_t, a_t)$ uses $\delta_i$ to estimate how much better the action $a_t$ is compared to the average action in state $s_t$. The critic is updated by minimising the temporal difference error:

$$\mathcal{L}_\delta = \mathbb{E}_{(s_t, a_t, r_t, s_{t+1}) \sim \rho_{\pi_i}} \left[ (r_t + \gamma \delta(s_{t+1}) - \delta(s_t))^2 \right], \tag{2}$$

where $\rho_{\pi_i}$ is the distribution induced by $\pi_i$. During policy execution, only the actor network is used.

Imitation learning (IL) is another solution method for MDPs. Instead of updating the policy using a loss derived from the rewards, the actor network is trained to imitate an expert policy. Behavioural Cloning (BC) is the simplest method of IL, which learns a policy directly from state-action pairs or action distributions gathered by executing the expert policy. If the expert policy's full action distribution $\pi_{\text{expert}}(\cdot \mid s)$ is available, the actor network is trained by minimising the Kullback-Leibler (KL) divergence between the predicted and expert distributions, given by $\mathcal{L}_{\pi KL}$. If only state-action pairs are available, the actor is trained by minimising cross-entropy loss $\mathcal{L}_{\pi CE}$:

$$\mathcal{L}_{\pi KL} = \mathbb{E}_{s \sim \rho_{\pi_{\text{expert}}}} \left[ D_{KL} \left( \pi_{\text{expert}}(\cdot \mid s) \parallel \pi(\cdot \mid s) \right) \right], \tag{3}$$

$$\mathcal{L}_{\pi CE} = -\mathbb{E}_{(s,a) \sim \rho_{\pi_{\text{expert}}}} \left[ \log \pi(a \mid s) \right]. \tag{4}$$

BC can also be used to pre-train an RL policy (Nair et al., 2018). In this case, the critic can be trained alongside the actor using the rewards from the environment using Equation (2).

## 4 Neural Algorithmic Reasoning as RL

Our method relies on modelling algorithms as MDPs, transforming the NAR problem from predicting output features to predicting a series of actions. GNARL is not a new RL algorithm in itself, but a framework that turns NAR problems into MDPs so that existing RL or IL algorithms, as well as tools such as action masking, can be used. The *Markov property*, requiring that future states depend only on the *current* state and action, straightforwardly holds for many polynomial algorithms, as was also recognised by Bohde et al. (2024). Similarly, the execution of a combinatorial optimisation process on a graph can often be framed as a sequence of choices among nodes or edges and represented using a GNN-based policy, as performed by the NCO and Graph RL works reviewed in Section 2.2. Using such an MDP framing, we unify the previously distinct paradigms of learning trajectories of algorithms for polynomial-time solvable and NP-hard combinatorial optimisation problems into a single task of learning a policy over graph element selections. This permits the use of the same learning architecture, which can be trained with or without an expert algorithm. As argued in Section 2.3, the purpose of this reformulation is to strengthen the general NAR approach, rather than to present GNARL as a replacement for task-specific NCO pipelines.

### 4.1 MDP Formulation

We define a graph algorithm MDP $\mathcal{M}_{\mathtt{A}}$ that models the execution of an algorithm $\mathtt{A}$ on a graph $G = (V, E)$. The algorithm operates over *input features* from an input space $\mathcal{I}$, which describe the problem instance, and *state features* from a feature space $\mathcal{F}$, which represent the algorithm's internal state, and are modified during execution. Additionally, each feature can be assigned to a *location*: node, edge, or graph. In NAR, $\mathcal{I}$ and $\mathcal{F}$ correspond to the input probes and hint probes, respectively. We assume that the output of the algorithm corresponds to a subset of the state features.

In general, a graph algorithm is easily framed as a sequential selection of nodes or edges, upon which an operation is applied. Aligning this concept with the MDP action, we consider the core action of $\mathcal{M}_{\mathtt{A}}$ to be the selection of a node $v \in V$. If a graph algorithm operates over edges, the edge can be constructed by selecting nodes in two consecutive *phases*. Thus, we define a graph feature $p$ in each state of $\mathcal{M}_{\mathtt{A}}$ which reflects the

current phase of the action selection. The maximum number of phases $\mathcal{P}$ is given by the problem, with values of 1, 2, and 3 corresponding to algorithms operating on nodes, edges, and triangles, respectively. To satisfy the Markov property, we additionally define the node state feature $\psi_p$, a one-hot feature which represents the node that was selected in phase $p$. Formally, we define $\mathcal{M}_\mathtt{A}$ as follows.

**States:** The states of $\mathcal{M}_\mathtt{A}$ correspond to the input features and state features: $S = \mathcal{I} \times \mathcal{F}$. This includes the phase feature $p$ and the previous node features $\psi_p$ required by the architecture, as well as any problem-specific features required to represent the algorithm.

**Actions:** The action space $\mathcal{A}$ is defined as the set of nodes $V$, and the actions available in each state $\mathcal{A}(s)$ are given by the problem according to simple rules.

**Transitions:** The transition function $\mathcal{T}$ is defined by the algorithm being executed, and generally corresponds to the fundamental internal update of the algorithm. Appendix C provides further discussion on defining the transition function.

**Rewards:** Suppose we wish to maximise an objective function $J : S \to \mathbb{R}$ in the terminal state of the MDP. We set the reward function $R = J(s') - J(s)$ for $s \in S \setminus s_0$, $R(s_0) = 0$. By the reward shaping theorem (Ng et al., 1999), the policy that maximises this reward function also maximises the MDP with the reward only in the terminal state.

**Horizon:** The maximum horizon $h$ is the worst-case number of steps for the algorithm. Unlike NAR, where the number of processor steps depends on the pre-determined quantity of hints or a separate termination network (Veličković et al., 2022), the horizon is independent of the trajectory.

As an example, we provide the MDP formulation for the DFS algorithm shown in Figure 1C, with others described in Appendix D. The input and state features can be found in Table 1, with types defined as per the CLRS-30 Benchmark. The features used are a simplification of the hints used in the Benchmark, with the addition of the phase and last selected features. The horizon $h = (|V| - 1)\mathcal{P}$. The transition function $\mathcal{T}$ is described in Algorithm 1, where $v$ is the selected action. The available actions are $\mathcal{A}(s) = V$ when $p = 1$ (all nodes), and $\mathcal{A}(s) = \{v | (\psi_1, v) \in E\}$ when $p = 2$ (outgoing edges of the previously selected node). For DFS, we train GNARL using IL only, and therefore a reward function is not required.

---
**Algorithm 1:** DFS Transition $\mathcal{T}$

---
$p \leftarrow 1$
**Function** STEPSTATE($v$)
    **if** $p = 2$ **then**
        $\mathtt{reach}_{\psi_1} \leftarrow 1$
        $\mathtt{reach}_v \leftarrow 1$
        $\mathtt{pred}_v \leftarrow \psi_1$
    $\psi_p \leftarrow v$
    $p \leftarrow p \bmod \mathcal{P} + 1$

---

Table 1: Features for the DFS algorithm.

| Feature | Description | Stage | Location | Type | Initial Value |
|---|---|---|---|---|---|
| `adj` | Adjacency matrix | Input | Edge | Scalar | - |
| $p$ | Phase ($\mathcal{P} = 2$) | State | Graph | Categorical | 1 |
| $\psi_m$ for $m = 1, \ldots, \mathcal{P}$ | Node last selected in phase $m$ | State | Node | Categorical | $\emptyset$ |
| `pred` | Predecessor in the tree | State | Node | Pointer | $v \forall v \in V$ |
| `reach` | Node has been searched | State | Node | Mask | $0 \forall v \in V$ |

## 4.2 Architecture

We replace the "encode-process-decode" paradigm from NAR with *encode-process-act*. The encode and process stages reflect their NAR counterparts, but in the final stage we transform the processed features into the *action probability* space instead of the input space.

**Encoder.** We employ an encoding process similar to Ibarz et al. (2022). At each step, for each distinct input and state feature, a linear transform maps the feature into the embedding space with dimension $f = 64$. The transformed features are then aggregated with other features of the same location (node, edge, or graph). This produces a graph encoding given by $\{\mathbf{z}_v^{(t)}, \mathbf{z}_{uv}^{(t)}, \mathbf{z}_g^{(t)}\}$, with shapes $n \times f$, $n \times n \times f$, and $f$ for node, edge, and graph features respectively. To maintain the Markov property, we encode all input and state features at every step.

**Processor.** The encoded features are fed into a GNN processor $P$, which performs $L$ rounds of message passing to calculate node embeddings $\mathbf{h}_v^{(t)}$. These node embeddings are passed to a pooling layer, which calculates an embedding for the graph $\overline{\mathbf{h}}^{(t)} = \text{pool}_{v \in V}(\mathbf{h}_v^{(t)})$. Bohde et al. (2024) find that removing the passing of latent encodings between NAR iterations achieves better algorithmic alignment with the Markov property and produces better performance. Thus, for the processor, we use a modified version of the MPNN in Veličković et al. (2022), in which the latent embeddings are removed, i.e., we omit $\mathbf{h}^{(t-1)}$ in Equation (1).

**Actor.** The actor network takes the node and graph embeddings as input and outputs a probability distribution over the actions. As the policy must be flexible w.r.t. graph size, we adapt the proto-action approach of Darvariu et al. (2021b). The proto-action is computed by applying a learned linear transform $\Theta$ to the graph embedding, further described in Appendix B.1. The similarity $\texttt{sim}$ between each node embedding and the proto-action is calculated using an Euclidean metric. We obtain a probability distribution over the actions using the softmax operator: $\pi(a_v|s) = \frac{\exp(\texttt{sim}_v/\zeta)}{\sum_{u \in V} \exp(\texttt{sim}_u/\zeta)}$, where $\zeta$ is a learned temperature and $\texttt{sim}_v = -\|\mathbf{h}_v^{(t)} - \Theta(\overline{\mathbf{h}}^{(t)})\|_2$.

### 4.3 Training

GNARL can be trained using both IL and RL. In problems with a clear algorithmic prior (e.g., the CLRS-30 Benchmark), we train using BC which closely parallels supervised learning, given the loss function $\mathcal{L}_\pi$ in Equations (3) and (4). Conversely, for many NP-hard problems, there may not be a clear expert to imitate, or the expert may not be able to scale to large enough problems to train the model. In this case, as long as an objective function is defined, training using RL means there is no reliance on an expert algorithm, overcoming a major limitation of existing NAR works. When training using RL, we use PPO (Schulman et al., 2017), which requires a critic module. For the critic network, we use an MLP which takes as input the graph embedding and outputs a scalar state value prediction. We train the critic as described in Section 3.2. When training using only BC, the critic module and reward function are not needed. However, if BC is used to pre-train a policy before RL fine-tuning, the critic can be warm-started by training it using Equation (2).

### 4.4 Validity, Correctness, Multiple Solutions, and Action Masking

A solution is considered *valid* if it satisfies the fundamental constraints specified by the problem. For example, for path-finding problems, a solution is valid if it does not violate the graph topology constraints (e.g., defining a predecessor edge where no edge exists). Validity also applies to combinatorial optimisation problems (e.g., a solution to the Travelling Salesperson Problem must be a cycle that visits each node exactly once).

We consider a solution to be *correct* if it can be produced by a deterministic algorithm with some fixed node ordering, aligning with Kujawa et al. (2025). Standard NAR requires a pre-defined node ordering (e.g., the `pos` feature in CLRS-30), and can only learn to produce the solution corresponding to that ordering. In contrast, GNARL learns a probability distribution $\pi$ over actions, and we omit any node ordering features. Thus, multiple correct solutions can be reached by *sampling* from the policy, or a deterministic solution can be found by selecting the highest-probability action at each step.

We note that correctness is a stricter requirement than validity. The GNARL framework uses action masking to prevent invalid selections outside of $\mathcal{A}(s)$ such as non-existent edges, guaranteeing that the solutions produced are valid by construction, but not necessarily correct, as the policy may not learn to select the correct actions. We do not use the correctness concept for combinatorial optimisation problems as we do not require the solution to be optimal, since this is often intractable for NP-hard problems.

### 4.5 Parallel Computation Trade-Offs

By framing a problem as an MDP, it becomes necessarily sequential, as the next state depends on the current state and action. In contrast, in the standard NAR framing, several sequential algorithmic steps can be represented using a single parallel step. In this work, we choose to encode the smallest, most fundamental dynamic update in the transition function, which precludes the parallel computation represented by a single NAR step. For instance, in the Bellman-Ford algorithm, a single NAR step represents the relaxation of all

Table 2: Solution correctness % on CLRS-30 Benchmark problems ($\uparrow$), $|V| = 64$, from 5 different seeds. Italicised results are estimates of graph accuracy (lower bound for BFS and DFS).

|  | TripletMPNN | Hint-ReLIC | G-ForgetNet | Kujawa et al. | DNAR | $\text{GNARL}_{\text{BC}}$ |
|---|---|---|---|---|---|---|
| BFS | $100.0_{\pm 0.0}$ | *52.6* | *97.5* | - | *100.0* | $100.0_{\pm 0.0}$ |
| DFS | $24.2_{\pm 28.6}$ | *100.0*$^*$ | *15.4* | 3 $_{\pm 0}$ | *100.0* | $100.0_{\pm 0.0}$ |
| Bellman-Ford | $12.4_{\pm 6.5}$ | *5.4* | *59.0* | 54 $_{\pm 20}$ | - | $93.6_{\pm 5.1}$ |
| MST-Prim | $2.6_{\pm 4.2}$ | *0.0* | *4.3* | - | *100.0* | $59.0_{\pm 10.8}$ |

edges from a node, whereas in GNARL we encode the transition function to be the relaxation of a single edge. While this avoids the policy becoming trivial, it follows that the number of computation steps required to execute the GNARL policy can be greater than those required by standard NAR. For further discussion of MDP design considerations, see Appendices C.2, C.4 and D.3.

## 5 Evaluation

In this section, we conduct our evaluation of GNARL over a series of classic graph problems with polynomial algorithms, NP-hard problems, and a problem lacking a strong expert. For brevity, many technical details and additional experiments are deferred to the Appendix.

### 5.1 CLRS Graph Problems

We first evaluate our approach on a selection of problems from the CLRS-30 Benchmark (Veličković et al., 2022) which are executed on an underlying graph. The chosen problems present varying levels of difficulty in baseline performance: BFS is consistently solved, while DFS results in highly mixed outcomes. Furthermore, Bellman-Ford and MST-Prim are solved with high node accuracy but poor graph accuracy by existing methods, and represent different underlying algorithmic structures. For the CLRS-30 problems, we train using BC based on the source algorithm, considering actions in a given state to be equally likely if the algorithm would produce them under different node orderings.

We report the percentage of correctly solved problems in Table 2, which clearly demonstrates the advantage of our approach compared to vanilla NAR (TripletMPNN). Of the baselines, only Kujawa et al. (2025) can produce multiple correct solutions, making it the most directly comparable method to our framework. Due to lack of code for reproduction, for Hint-ReLIC (Bevilacqua et al., 2023) and (G-)ForgetNet (Bohde et al., 2024), we report an estimate of the graph accuracy, rather than solution correctness, using `node_accuracy`$^{|V|}$. For Bellman-Ford and MST-Prim, graph accuracy is equivalent to solution correctness, while for DFS and BFS, graph accuracy is a strict lower bound on solution correctness due to the possible existence of multiple correct solutions. We provide further analysis in Appendix G.1, and conclude that the approximation does not significantly affect the trend of the results, which is that GNARL matches or outperforms the baselines. Hint-ReLIC makes use of additional hints with reversed pointers which enables 100% OOD node accuracy on DFS, while we use only the original pointers from CLRS-30. Several problems from CLRS-30 have been solved with 100% correctness by DNAR (Rodionov & Prokhorenkova, 2025), which uses separate discrete and scalar operators, closely mirroring the operations of classic algorithms using teacher forcing. While this enables perfect performance in the P-hard CLRS-30 problems, we show in Appendix G.2 that this approach does not generalise well to combinatorial optimisation problems where approximations are required to maintain polynomial complexity.

#### 5.1.1 Finding Multiple Solutions

Using the action probabilities output by GNARL, we can explore the ability of the model to find multiple correct solutions for problems with non-unique solutions, such as BFS and DFS. Given the distribution, we sample an action at each step with a temperature parameter $\lambda$: $\pi_\lambda(a|s) \propto \pi(a|s)^{1/\lambda}$. When $\lambda \to 0$, the action with the highest probability is always chosen, as in Table 2, while when $\lambda \to \infty$ actions are chosen uniformly at random from the available actions. Figure 2 demonstrates the number of unique solutions found

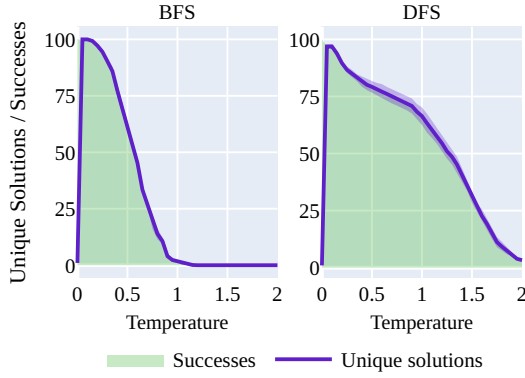

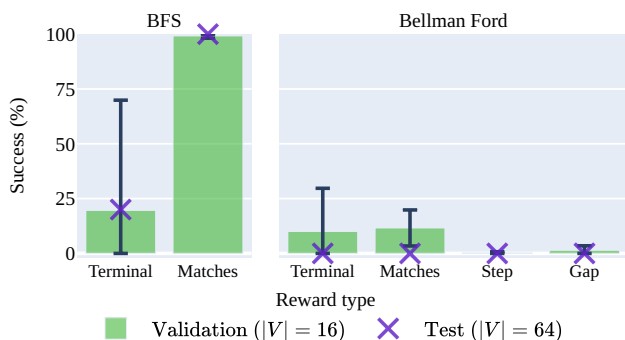

Figure 2: Unique solutions in 100 runs found on a single graph, $|V| = 64$, using sampled actions (10 seeds).

Figure 3: PPO performance on BFS and Bellman-Ford under different reward functions (5 seeds, $10^7$ steps).

for BFS and DFS on a single graph by sampling actions from the trained models for 100 episodes. Even with temperatures very close to zero, all the solutions found are unique. For $\lambda = 0.05$, BFS finds 100 unique correct solutions, while DFS has a 96.9% success rate with each solution being unique. With higher temperatures, the success rate decreases, as more incorrect actions are sampled.

### 5.1.2 Solving CLRS-30 Problems with RL

While the CLRS-30 Benchmark is designed for supervised learning and does not naturally define objective functions, we may attempt to solve the problems using RL by designing a reward function in order to showcase the flexibility of GNARL. As designing an appropriate reward function is a well-known challenge in the RL literature (Booth et al., 2023), we base the objective functions on simple metrics. For the BFS and Bellman-Ford problems, we define an objective function based on the *matches* of true path distances: $J_{\text{match}}(s) = \frac{1}{|\mathcal{V}|} \left( \sum_{v \in \mathcal{V}} \mathbb{1}\left[f_s(v) = f^*(v)\right] \right)$, where $f(v)$ is the length of the path of node $v$ to the source in the predecessor tree, and $f^*$ is the reference solution. For Bellman-Ford, we may also define a *gap-based* objective: $J_{\text{gap}}(s) = \frac{1}{|\mathcal{V}|} \left( \sum_{v \in \mathcal{V}} f^*(v) - f_s(v) \right)$. We alternatively consider a *terminal* reward of $+1$ for a correct solution, and a *step*-based reward of $-1$ per step until termination.

Figure 3 shows the performance of PPO using different reward functions on the BFS and Bellman-Ford problems. For BFS, PPO achieves a perfect success rate under the *matches* objective, though the model is much more sensitive to hyperparameter choice than when trained with BC. The *terminal* reward exhibits high variance, with one seed achieving 100% test success while the others achieve 0%. For Bellman-Ford, no model seeds achieve a validation success rate greater than 40%. While the *matches* objective yields the best average performance for these environments, it is not possible to define it analogously for non-path-finding problems, like DFS and MST-Prim. Overall, we demonstrate that, while it is possible to apply RL to CLRS-30 problems, the challenge of defining a suitable reward function in the absence of an objective function makes it difficult to achieve good performance, and IL ought to be preferred when expert trajectories are available.

The CLRS-30 Benchmark problems are a useful tool to evaluate the reasoning capabilities of GNARL against established baselines. However, to date, NAR methods have struggled to extend to combinatorial optimisation problems, which are of greater practical interest. The following sections of our evaluation demonstrate the ability of GNARL, despite its general architecture, to outperform prior NAR methods dedicated to specific combinatorial optimisation problems.

### 5.2 Minimum Vertex Cover

The Minimum Vertex Cover (MVC) is a classic NP-hard problem with broad applicability, and has been approached using NAR in He & Vitercik (2025). Their PDNAR method uses a dual problem formulation to create a bipartite graph for training using both an approximation algorithm and the optimal solution, followed

Table 3: Model-to-algorithm ratio of objective functions ($J/J_{\text{approx}}$) for MVC ($\downarrow$). Models trained directly on the approximation algorithm steps are italicised. Averaged from 5 different seeds.

| | Test size | | | | | | |
|---|---|---|---|---|---|---|---|
| $J/J_{\text{approx}}$ | 16 (1×) | 32 (2×) | 64 (4×) | 128 (8×) | 256 (16×) | 512 (32×) | 1024 (64×) |
| *PDNAR* | *0.943* $_{\pm 0.004}$ | *0.957* $_{\pm 0.002}$ | *0.966* $_{\pm 0.002}$ | *0.958* $_{\pm 0.002}$ | *0.958* $_{\pm 0.002}$ | *0.958* $_{\pm 0.002}$ | *0.957* $_{\pm 0.002}$ |
| PDNAR$_{\text{No algo}}$ | 1.142 $_{\pm 0.038}$ | 1.115 $_{\pm 0.027}$ | 1.110 $_{\pm 0.038}$ | 1.099 $_{\pm 0.032}$ | 1.091 $_{\pm 0.034}$ | 1.099 $_{\pm 0.036}$ | 1.095 $_{\pm 0.038}$ |
| GNARL$_{\text{BC}}$ | 0.9398 $_{\pm 0.0008}$ | 0.9420 $_{\pm 0.0007}$ | 0.9478 $_{\pm 0.0011}$ | 0.9463 $_{\pm 0.0016}$ | 0.9500 $_{\pm 0.0019}$ | 0.9496 $_{\pm 0.0029}$ | 0.9497 $_{\pm 0.0025}$ |
| GNARL$_{\text{PPO}}$ | **0.9391**$_{\pm 0.0003}$ | **0.9413**$_{\pm 0.0005}$ | **0.9448**$_{\pm 0.0005}$ | **0.9428**$_{\pm 0.0006}$ | **0.9462**$_{\pm 0.0003}$ | **0.9449**$_{\pm 0.0003}$ | **0.9453**$_{\pm 0.0004}$ |

by a solution clean-up stage. We model the MVC MDP as a sequential selection of nodes comprising the vertex cover. We create valid-by-construction solutions using the action validity function $\mathcal{A}(s) = \{v \mid \texttt{in\_cover}_v = 0\}$, removing the need for a clean-up stage. As a baseline, we compute the $2/(1 - \epsilon)$-approximation from Khuller et al. (1994), with $\epsilon = 0.1$. We use the training data from He & Vitercik (2025), consisting of $10^3$ Barabási-Albert (BA) graphs with $|V| = 16$, and we validate and test on their data. For BC, we train using expert demonstrations generated by an exact ILP solver, and for PPO, we train directly using the reward function, *without* any expert data.

Results in Table 3 show the performance of different models as a ratio of the performance of the approximation algorithm. We include PDNAR$_{\text{No algo}}$, which is an ablation of PDNAR using only information from the optimal solution, the same data provided to GNARL. GNARL trained on expert trajectories outperforms all baselines, including the full PDNAR model, across all graph sizes, without relying on any approximation algorithm trajectories or cleanup stages. Remarkably, GNARL trained with only a reward signal achieves the best performance out of all methods, even at scales 64× larger than the training size, significantly improving on the approximation algorithm and the PDNAR baselines. Our general method achieves state-of-the-art performance among NAR methods on the MVC problem without requiring a dual formulation, approximation algorithm, or refinement stage, greatly simplifying both the modelling and solution pipelines.

### 5.3 Travelling Salesperson Problem (TSP)

The TSP is a thoroughly studied combinatorial optimisation problem, allowing us to evaluate the performance of GNARL with an optimal baseline. The NAR approach of Georgiev et al. (2024a) predicts the probability of each node being a node's predecessor in a tour, then extracts valid tours using beam search. We demonstrate that we can produce valid-by-construction solutions, removing the need for beam search, and achieve superior performance using both IL and RL as training methods. This comparison is included to show the benefit of the proposed reformulation against existing NAR approaches on the TSP, not to claim that GNARL is a general-purpose NCO competitor, as argued in Section 2.3.

We model the TSP using a constructive approach in which the action adds the node to the tour, terminating when all nodes have been selected. Actions are restricted to $\mathcal{A}(s) = \{v \mid \texttt{in\_tour}_v = 0\}$, thus all solutions are valid by construction. We use the same input features as Georgiev et al. (2024a), and train using 10% of their training data. We first train a policy using BC, with demonstrations generated from the optimal solutions provided by the Concorde solver (Applegate et al., 1998). We also train a policy via PPO, relying only on the optimisation of the reward function.

Results in Table 4 show the performance of GNARL trained with BC and GNARL trained with PPO as compared to the *best* results of all methods presented by Georgiev et al. (2024a), and against the handcrafted Christofides heuristic (Christofides, 1976). These results demonstrate that GNARL trained on expert trajectories in small graphs is able to generalise effectively to much larger graphs, significantly outperforming both the heuristic and the prior NAR approach, despite using a fraction of the training data and not using beam search. GNARL trained with only PPO is also able to achieve strong performance in graphs up to 5× the training size, but, interestingly, does not generalise to OOD graph sizes as well as GNARL trained with BC. Further investigations on pre-training using a weak expert can be found in Appendix G.3.

Our tour construction process aligns with the only other existing NAR work addressing the TSP to enable a fair comparison. However, other modelling choices may lead to better approximation ratios. For example,

Table 4: TSP percentage worse than optimal objective for OOD graph sizes (↓). Non-NAR baselines are italicised. Additional baselines can be found in Georgiev et al. (2024a).

| Model | Test size | | | | | |
|---|---|---|---|---|---|---|
| | 40 (2×) | 60 (3×) | 80 (4×) | 100 (5×) | 200 (10×) | 1000 (50×) |
| *Christofides* | *10.1$_{\pm 3}$* | *11.0$_{\pm 2}$* | *11.3$_{\pm 2}$* | *12.1$_{\pm 2}$* | *12.2$_{\pm 1}$* | *12.2$_{\pm 0.1}$* |
| Georgiev et al. | 7.5$_{\pm 1}$ | 13.3$_{\pm 3}$ | 19.0$_{\pm 3}$ | 23.8$_{\pm 5}$ | 33.6$_{\pm 3}$ | 38.5$_{\pm 3}$ |
| GNARL$_{BC}$ | **2.2**$_{\pm 0.1}$ | **3.6**$_{\pm 0.2}$ | **3.9**$_{\pm 0.4}$ | **4.4**$_{\pm 0.2}$ | **6.4**$_{\pm 0.6}$ | **11.8**$_{\pm 1.4}$ |
| GNARL$_{PPO}$ | 3.0$_{\pm 0.3}$ | 4.9$_{\pm 0.5}$ | 7.1$_{\pm 1.2}$ | 8.6$_{\pm 1.5}$ | 16.7$_{\pm 4.5}$ | 45.2$_{\pm 15.4}$ |
| GNARL$_{PPO+heur}$ | 5.3$_{\pm 0.6}$ | 8.3$_{\pm 0.9}$ | 9.8$_{\pm 1.6}$ | 11.3$_{\pm 1.7}$ | 15.4$_{\pm 2.2}$ | 20.7$_{\pm 2.6}$ |

Table 5: Inference time per graph (in seconds) for GNARL and NAR models on TSP test data.

| | | Georgiev et al. (total) | Georgiev et al. (beam search) | GNARL (total) | GNARL (environment) |
|---|---|---|---|---|---|
| | 40 | 1.17$_{\pm 0.02}$ | 0.553$_{\pm 0.012}$ | 0.798$_{\pm 0.004}$ | 0.0428$_{\pm 0.0041}$ |
| | 60 | 3.95$_{\pm 0.14}$ | 1.92$_{\pm 0.09}$ | 2.58$_{\pm 0.02}$ | 0.0660$_{\pm 0.0059}$ |
| | 80 | 9.94$_{\pm 0.25}$ | 5.01$_{\pm 0.16}$ | 5.99$_{\pm 0.03}$ | 0.103$_{\pm 0.006}$ |
| **Test size** | 100 | 18.7$_{\pm 0.5}$ | 9.26$_{\pm 0.36}$ | 12.5$_{\pm 0.1}$ | 0.163$_{\pm 0.005}$ |
| | 200 | 149$_{\pm 5}$ | 73.4$_{\pm 3.4}$ | 98.1$_{\pm 0.6}$ | 0.698$_{\pm 0.022}$ |
| | 1000 | 12800$_{\pm 171}$ | 2140$_{\pm 15}$ | 15000$_{\pm 87}$ | 67.7$_{\pm 1.0}$ |

in the NCO literature, Khalil et al. (2017) use a helper function in $\mathcal{T}$ that determines the best insertion position for a node instead of mandating insertions be made at the head. GNARL is sufficiently flexible to allow different MDP models, and we show the result of using this insertion heuristic in the table (PPO+heur). Interestingly, the heuristic improves the performance of PPO-trained GNARL at larger scales, but does not perform as well on graphs closer to the size of the training data.

### 5.3.1 Inference Time Comparison

Table 5 shows the inference time per graph on the test data for the TSP, using GNARL and the MPNN model from Georgiev et al. (2024a). For the Georgiev et al. model, we use a beam search width of 1280, as this is the width used to achieve the results reported in Table 4. The results were obtained on a single core of an Intel Platinum 8628 CPU, and are the average over 5 models. For the GNARL model, inference on graphs of size 1000 required 20GB of memory, while the Georgiev et al. model required 64GB for the same size. Here, a direct comparison is possible as both models are implemented in the same framework.

The results demonstrate that GNARL achieves statistically significantly faster inference times for most problem sizes, due to the lack of reliance on beam search. The beam search uses approximately half of the total inference time of the NAR approach at each size. In comparison, the environment step of the GNARL model requires only a small fraction of the total inference time, making the overall approach faster despite the similar amount of time required for the model's forward pass. For the largest problem size of 1000 nodes, the long episode length required for the sequential decision-making process overtakes the advantage of not using beam search, leading to a slightly longer inference time for GNARL.

### 5.4 Robust Graph Construction

We also study the robust graph construction (RGC) domain from Darvariu et al. (2021a), in which edges are added to a graph to improve the robustness against disconnection under node removal. In this problem there is no clear expert, and the objective function itself is expensive to calculate, meaning that a neural approach is highly appropriate. Problems of this nature are common in many practical contexts, yet have not previously been approached in the NAR literature.

We use data from Darvariu et al. (2021a) to evaluate the performance of GNARL in contrast to their purpose-built graph RL model, RNet-DQN. We design state features and the transition function

similarly to RNet-DQN, but in a way which aligns with the GNARL framework. All models are trained on a set of Erdős-Renyi (ER) or BA graphs with $|V| = 20$, with the removal strategy being either random or targeted as per Darvariu et al. As there is no expert algorithm for this problem, we train using PPO directly for $10^7$ steps. We also investigate the effect of warm-starting training using BC: we consider a weak expert (WE) policy which greedily selects the next edge, and fine-tune using PPO (WE+PPO).

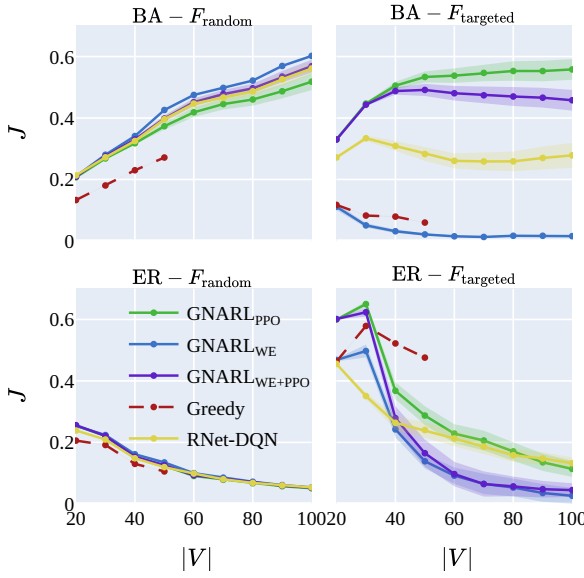

Figure 4: Performance on RGC for different sizes (↑).

Results in Figure 4 (detailed in Table 14 in the Appendix together with uncertainty estimates) show the performance of each method on different OOD graph sizes. The GNARL approaches are competitive with RNet-DQN, which is expected due to the similarities in architecture. Interestingly, for BA graphs under the targeted removal strategy, GNARL trained with PPO improves on the generalisation ability of RNet-DQN, while the policy trained only on the weak expert performs very poorly on this class of graphs. While a traditional NAR approach is not applicable in this scenario due to the absence of an expert algorithm, GNARL allows us to solve this problem using a specification very similar to those used in NAR.

## 6  Limitations and Future Work

There are several important directions for future work. First, defining the MDP presently requires knowledge of the basic steps for constructing a valid solution. In contrast with standard NAR, which can be executed without hints at runtime, it is not possible to execute GNARL without an MDP definition. As with the design of NAR hints, GNARL also requires practitioners to encode expertise in the design of the MDP. Furthermore, GNARL relies on the environment during execution, creating a performance bottleneck. Both of these limitations might be addressed by using learnable world models (Schrittwieser et al., 2020; Chung et al., 2023) instead of explicitly defined transition functions. Second, when training a model using BC, states outside of $\rho_{\pi_{\text{expert}}}$ are not encountered. During execution, if a sampled action causes a transition to a state outside of $\rho_{\pi_{\text{expert}}}$, the model is not likely to correctly predict the best actions, and the episode often ends in failure. For longer action sequences, the chance of encountering such an action is higher, and as a result the model does not perform as well in problems with longer trajectories. In future work, this could be mitigated using a more advanced imitation learning technique such as DITTO (DeMoss et al., 2025), which brings expert trajectories and policy rollouts closer together in the latent space of a learned world model. This offers a principled and promising way of improving the robustness of GNARL.

## 7  Conclusion

In this work, we have presented GNARL, a framework that reimagines the learning of algorithmic trajectories as Markov Decision Processes. Doing so unlocks the powerful sequential decision-making tools of reinforcement learning for NAR and serves as a critical bridge between the two largely distinct fields. GNARL addresses the key limitations of classic NAR and is broadly applicable to problems spanning those solvable by polynomial algorithms, as well as challenging combinatorial optimisation problems. Our approach can natively represent multiple correct solutions, removes the need for inference-time repair procedures, and performs on par with or better than much narrower NAR methods on NP-hard problems. Furthermore, despite occasional poorer stability and generalisability when using reward-driven learning, GNARL can be applied even in situations where an expert algorithm is missing entirely. In our view, this work is an important step towards achieving a combinatorial optimisation framework that abstracts away from technical details, as envisaged by Cappart et al. (2023).

**Statement of Broader Impact**

This work is primarily methodological, and we do not foresee any direct societal consequences. However, the application of learned combinatorial optimisation policies to areas such as infrastructure, logistics, or network design may have negative operational consequences if distribution shifts occur. The framework currently provides no guarantees about optimality or robustness outside of the evaluated distributions. While GNARL helps enforce solution validity, performance degradation (in terms of correctness or optimality gap) can still be substantial. Furthermore, the computational cost of training and executing learned policies may be higher than conventional algorithms and heuristics, which should be considered in future applications of GNARL.

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

# A  Reproducibility Statement

Code required to reproduce the results of the paper will be released upon publication. Experiments were implemented using PyTorch Geometric (Fey & Lenssen, 2019), Gymnasium (Towers et al., 2024), Stable Baselines 3 (Raffin et al., 2021) libraries. The MPNN implementation was adapted from Georgiev et al. (2024b). Each training run was executed on a single core of an Intel Platinum 8628 CPU with 4GB of memory using CentOS Linux 8.1.

# B  Architecture Details

Figure 5 shows the architecture of the GNARL framework, with each stage highlighted. This represents a much more detailed diagram than the summary provided in Figure 1B.

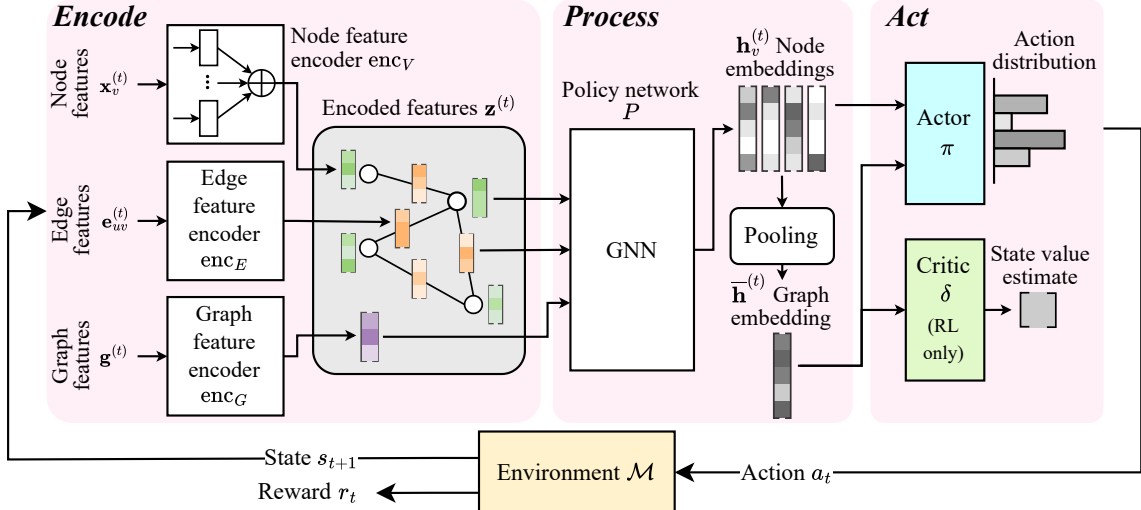

Figure 5: Architecture of the GNARL framework. Each state and input feature is encoded separately, and aggregated by feature location. The processor embeds the state features into a proto-action space. The actor calculates node probabilities using the similarity of the graph embedding to the proto-action vectors. The critic uses the graph embedding to estimate the state value.

## B.1  Actor Network

Further details of the actor network, including the proto-action mechanism, are provided in Figure 6. The action distribution is computed by comparing each node embedding to a proto-action vector, which represents the desired characteristics of the next node to be selected. The graph embedding is passed to a learned linear transformation to produce the proto-action. This proto-action is then compared to the embedding of each node using a similarity function, in this case negative Euclidean distance. The action distribution is produced by passing the similarity scores through a softmax function with a learned temperature.

# C  Obtaining an MDP from an Algorithm

## C.1  Worked Example

The goal of the GNARL framework is to learn to execute classic algorithms by modelling them as MDPs. In order to model a classic algorithm as an MDP, we must define a transition function, which represents the way in which the algorithm's state changes when an action is taken. Designing the MDP based on the algorithm is a matter of identifying the decision variables, which become the actions, and the state updates, which become the transition function. We illustrate this process using the Bellman-Ford algorithm, given in

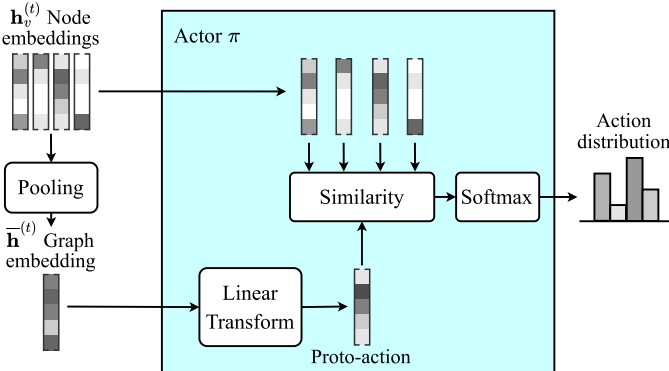

Figure 6: The proto-action is generated from a linear transformation of the graph embedding. This proto-action is then compared to each node embedding using Euclidean distance, which is negated and passed through a softmax function to produce the action distribution.

Algorithm 2, as defined by the CLRS-30 Benchmark (Veličković et al., 2022). The input, output, and hint features used in the CLRS-30 Benchmark for Bellman-Ford are given in Table 6.

---

**Algorithm 2:** Bellman-Ford Reference Algorithm

---

**Input:** Adjacency matrix `adj`, weight matrix $A \in \mathbb{R}^{n \times n}$, start node $v_s$
**Output:** Predecessor array `pred` $\in \{1, \ldots, n\}^n$

**1** Obtain $G = (V, E)$ from `adj`
**2** $d_v \leftarrow 0 \ \forall v \in V$
**3** $\texttt{pred}_v \leftarrow v \ \forall v \in V$
**4** $\texttt{mask}_v \leftarrow 0 \ \forall v \in V$
**5** $\texttt{mask}_{v_s} \leftarrow 1$

**6** **while** True **do**
**7** $\quad$ $d_{\mathrm{prev}} \leftarrow d$
**8** $\quad$ $\texttt{mask}_{\mathrm{prev}} \leftarrow \texttt{mask}$
**9** $\quad$ **foreach** $u \in V$ **do**
**10** $\quad\quad$ **foreach** $v \in V$ **do**
**11** $\quad\quad\quad$ **if** $\texttt{mask}_{\mathrm{prev}_u} = 1$ **and** $(u, v) \in E$ **then**
**12** $\quad\quad\quad\quad$ **if** $\texttt{mask}_v = 0$ **or** $d_{\mathrm{prev}_u} + A_{u,v} < d_v$ **then**
**13** $\quad\quad\quad\quad\quad$ $d_v \leftarrow d_{\mathrm{prev}_u} + A_{u,v}$
**14** $\quad\quad\quad\quad\quad$ $\texttt{pred}_v \leftarrow u$
**15** $\quad\quad\quad\quad$ $\texttt{mask}_v \leftarrow 1$

**16** $\quad$ **if** $d = d_{\mathrm{prev}}$ **then**
**17** $\quad\quad$ **break**

**18** **return** `pred`

---

The Bellman-Ford algorithm consists of a main loop (Lines 6-17) which continues until no distance updates occur (Line 16). Within this loop, each edge outgoing from visited nodes is relaxed (Lines 11-15). Thus, the fundamental unit of the algorithm is to relax a given edge. We can therefore represent the MDP such that the transition function pertains to the relaxation of a single edge. As the GNARL framework operates over nodes rather than edges, we split the selection of an edge $(u, v)$ into two steps: first selecting the source node $u$, then selecting the target node $v$. Thus, we choose $\mathcal{P} = 2$, and using the features from the CLRS-30 Benchmark, arrive at the state features described in Table 7. Note that the adjacency matrix is not listed as an explicit feature as it can be inferred from the non-zero entries in the weight matrix, $A$.

Table 6: Features in the CLRS-30 Benchmark for the Bellman-Ford algorithm

| Feature | Description | Stage | Location | Type | Initial Value |
|---------|-------------|-------|----------|------|---------------|
| adj | Adjacency matrix | Input | Edge | Scalar | Given |
| $A$ | Weight matrix | Input | Edge | Scalar | Given |
| $v_s$ | Start node | Input | Node | Mask One | Given |
| pred | Predecessor in the shortest path | Hint/Output | Node | Pointer | $v \ \forall v \in V$ |
| mask | Node has been visited | Hint | Node | Mask | $0 \ \forall v \in V$ |
| $d$ | Current best distance | Hint | Node | Float | $0 \ \forall v \in V$ |

Table 7: Features for the Bellman-Ford algorithm.

| Feature | Description | Stage | Location | Type | Initial Value |
|---------|-------------|-------|----------|------|---------------|
| $A$ | Weight matrix | Input | Edge | Scalar | - |
| $v_s$ | Start node | Input | Node | Mask One | - |
| $p$ | Phase ($\mathcal{P} = 2$) | State | Graph | Categorical | 1 |
| $\psi_m$ for $m = 1, \dots, \mathcal{P}$ | Node selected in phase $m$ | State | Node | Categorical | $\emptyset$ |
| pred | Predecessor in the shortest path | State | Node | Pointer | $v \ \forall v \in V$ |
| mask | Node has been visited | State | Node | Mask | $0 \ \forall v \in V$ |
| $d$ | Current best distance | State | Node | Float | $0 \ \forall v \in V$ |

All that remains is to define the transition function that operates on the defined state features. This is given by the edge relaxation update in Lines 13-15. Notably, we do not include the conditional checks from Lines 11 and 16 in the transition function, as these must be learned by the model. The resultant MDP transition function is given in Algorithm 3. In this transition function, the edge relaxation (Lines 4-6) is only performed in the second phase (Line 2), after both the source node and target node have been selected. To guard against invalid actions, we require that the edge being relaxed exists in the graph (Line 3). Lines 7 and 8 update the feature for the previously selected node and the current phase respectively, as required to keep the Markov property. Using this process, we have derived an MDP representation of the Bellman-Ford algorithm from its classic definition.

---

**Algorithm 3:** Bellman-Ford Transition Function $\mathcal{T}$

---

**1 Function** STEPSTATE($v$)
**2**    **if** $p = 2$ **then**
**3**      **if** $(\psi_1, v) \in E$ **then**
**4**        $d_v \leftarrow d_{\psi_1} + A_{\psi_1, v}$
**5**        $\text{pred}_v \leftarrow \psi_1$
**6**        $\text{mask}_v \leftarrow 1$
**7**    $\psi_p \leftarrow v$
**8**    $p \leftarrow (p \bmod \mathcal{P}) + 1$

---

### C.2 MDP Design Considerations

In designing the MDP, some considerations must be made for the representation of the algorithm. The choice of representation is not unique, and different design decisions may lead to different MDPs for the same algorithm. In the case of Bellman-Ford, we made the design decision to have single-edge relaxations as the fundamental unit of the MDP. An alternative design could have used the choice of node $u$ from Algorithm 2 Line 9 as the action, and had the transition function relax all outgoing edges from $u$. This would greatly reduce the number of steps required to execute the algorithm, as the increased parallelism would allow multiple edges to be relaxed in a single step. However, this would also increase the complexity of the transition function, as it would need to perform multiple edge relaxations and conditional checks. As the goal

is for the model to learn the algorithm, this choice of MDP was not used, as it would offload much of the algorithm's logic to the transition function rather than the model. In this manner, the design of the MDP can affect the difficulty of the learning task.

Another important design consideration is maintaining the Markov property of the MDP. This requires that the state features contain all information necessary to determine the next state given an action. As algorithms operate on their internal state, it should always be possible to define a Markovian representation of the algorithm, being careful to include all relevant algorithm state information. Generally, care should be taken when defining a transition function for an algorithm which relies on implicit ordering information, such as for loops, or derived information, like the root node of a tree, and it may be necessary to include additional state features to capture this information.

The requirement of a hand-designed MDP representation is a limitation of the GNARL framework, although we view it as similar to the hand-selection of features used in standard NAR architectures (Veličković et al., 2022). In future work, we hope to address this limitation by using world modelling techniques to learn the MDP representation directly from the algorithm's execution trace.

## C.3   Termination

The termination condition of the MDP is defined by the horizon $h$, which is assumed to be provided in the problem definition and is typically a simple function of the number of nodes in the graph. The horizon acts as an upper bound on the number of steps required to reach a solution. Therefore, in algorithmic environments, the horizon is set based on the worst-case time complexity of the algorithm.

In the CLRS-30 Benchmark, the number of steps in an evaluated trajectory is pre-determined by the reference algorithm and is considered an input to the system. As such, we adopt an analogous practice in GNARL, where a trajectory may optionally terminate once a correct solution is reached. The algorithm thus requires one solution as a reference to determine whether this early termination condition is met, although any solution can be found by the algorithm and may not be equal to the reference solution. While this does not reflect a deployable procedure in practice, we consider that requiring an existing solution is feasible for the CLRS-30 problems, since these problems are solvable using algorithms of polynomial complexity. We employ this type of early termination in Bellman-Ford and MST-Prim (as DFS and BFS have a fixed number of steps).

In general, GNARL relies only on the horizon for termination, and no reference solution is used for this purpose when solving the combinatorial optimisation problems.

## C.4   MDP Steps vs NAR Steps

A key point of difference between the NAR and GNARL representations of an algorithm is the level of parallelism occurring in each execution step. In a single NAR step, the labels for every node and edge are predicted, which allows multiple updates to occur simultaneously. Conversely in GNARL, each step corresponds to a single node selection action, which allows for generality in modelling but results in less parallelism. For example, in the CLRS-30 Benchmark, the Bellman-Ford algorithm is highly parallelised so that each step represents a relaxation of every edge in the graph. In contrast, the GNARL implementation of Bellman-Ford relaxes a single edge for every $\mathcal{P} = 2$ steps, leading to a much longer episode length. As discussed in Appendix C.2, this design decision was made to ensure that the entire algorithm process was learned by the model, rather than relying on a complex transition function to perform the majority of the work.

For BFS and DFS, the episode length is fixed at $h$, while for Bellman-Ford and MST-Prim, the episode length may be shorter if the solution is reached early. On the MST-Prim test set, the expert achieves an average length of 462.84, while the horizon is 8192. For Bellman-Ford, the worst-case complexity is $\mathcal{O}(|V|^3)$, so we set the horizon of each problem to be twice the trajectory length achieved by the expert policy, to allow faster evaluation. On the Bellman-Ford test set, the expert achieves an average length of 446.6.

Table 8 shows the number of steps taken by the NAR and GNARL models for each algorithm on graphs with $|V| = 64$, averaged over 5 models on the test set of 100 graphs. Clearly, while the design decision to use

single-node updates in GNARL allows for a simple and general transition function, it comes at the cost of a much larger number of steps compared to NAR. This is exacerbated by models with lower success rates, such as MST-Prim, where the generous horizon means that unsuccessful episodes greatly skew the average step count. Interestingly, the number of steps taken by GNARL in successful episodes of Bellman-Ford is lower than the expert's average step count, indicating that the learned model is able to find solutions more efficiently than the expert algorithm in some cases. The impact on inference time is discussed in Appendix F.1. In future work, we intend to address this limitation by exploring methods of modelling parallel action selection in the MDP.

Table 8: Number of steps taken for NAR and GNARL models on graphs with $|V| = 64$.

| Method | BFS | DFS | Bellman-Ford | MST-Prim |
|---|---|---|---|---|
| NAR | $3.39_{\pm 0}$ | $192_{\pm 0}$ | $6.41_{\pm 0}$ | $65_{\pm 0}$ |
| GNARL | $128_{\pm 0}$ | $128_{\pm 0}$ | $475_{\pm 127}$ | $3630_{\pm 830}$ |
| GNARL (success only) | $128_{\pm 0}$ | $128_{\pm 0}$ | $400_{\pm 30}$ | $462_{\pm 1}$ |
| GNARL (failure only) | $128_{\pm 0}$ | $128_{\pm 0}$ | $923_{\pm 59}$ | $8192_{\pm 0}$ |

# D  Environment Details

## D.1  Features, Types, and Stages

The GNARL implementation framework inherits many aspects of its specification from the CLRS-30 Benchmark (Veličković et al., 2022). Features are the variables serving as the working space of an algorithm. In the benchmark, the state of the features at a given step is called a probe. Each probe has a stage, location, and type. The location is one of node, edge, or graph. The type defines how the probe is represented, and the CLRS-30 Benchmark defines various possible types, such as scalar, categorical, mask, mask-one, and pointer. In the benchmark, each type corresponds to a different loss function, but this does not apply for GNARL as training is performed on either the action distribution or the state-action-reward tuple. Each probe is initialised as per the CLRS-30 Benchmark unless otherwise stated.

In the CLRS-30 Benchmark, the stage of a probe can be either input, hint, or output. Inputs are encoded once in the first round of inference, while hints act as auxiliary probes for intermediate predictions, trained against reference hints. The output probes are trained separately and use their own loss function. We note that hint probes often represent intermediate values of the outputs.

The GNARL framework uses stages in a slightly different manner. There are two stage types: input and state. The input features correspond to immutable features of the graph that are given to the algorithm, and are sufficient for the expert algorithm to solve the problem. This corresponds to the CLRS-30 input features. Unlike NAR, we encode the input features at every step rather than just the first, to ensure that the Markov property holds in the absence of recurrent features in the architecture. In GNARL, we use state features to denote features which may be altered by the transition function during an episode. This is analogous to the hints in NAR. At each step during execution, both the input features and state features are encoded. We do not use a distinct output feature, and instead consider the output of the algorithm to be some subset of the state features. The evaluation of GNARL relies on the terminal state and/or the reward function, so there is no need to use a separate output feature.

## D.2  Implementation Note

Following Minder et al. (2023) and Georgiev et al. (2024b), we use a sparse implementation of the MPNN processor, where messages are only passed along existing edges in the graph. This allows scalability to large graphs, but restricts features to being defined on existing edges only, meaning that not all CLRS-30 graph problems are representable without additional engineering. This is a property of the implementation rather than the GNARL framework itself.

## D.3 Encoding Knowledge in the MDP and Role of the Policy

In NAR, the neural architecture is responsible for learning the algorithmic update at each step based on data from the step updates of the reference algorithm. In this case, the decision on where to place the step boundaries is made by the designer. In GNARL, the neural policy is primarily responsible for discovering the order of node selections within an algorithmic trajectory, while the underlying deterministic update primitive is encoded in the MDP transition function. For example, in the design we used for Bellman-Ford, the policy is responsible for selecting the next edge to relax, while the transition function performs the update mechanism to relax the edge. The order of node selection remains a challenging task, as decisions at a particular timestep can have ramifications much later in the episode.

Conversely, if the basic problem or algorithmic structure is already known, our view is that it should be encoded in the MDP rather than learned. Simple rules, such as adding a node to a set, or only selecting edges that exist, can easily be encoded in the transition function or action masks, allowing the neural policy to focus on the more complex task of learning the order of node selections. As demonstrated by the empirical results, this format allows much more reliable recreation of algorithmic trajectories, and in some cases even shows the promise to improve upon the reference algorithm (successful trajectories in Bellman-Ford have fewer steps than the reference algorithm, as shown in Table 8).

In the designs of the environments, which are detailed in the rest of this section, we have used the simplest MDP that can represent the problem while maximising the role of the policy in learning the algorithmic trajectory. We expressly avoided encoding the rules used by the algorithm for next node selection in the action mask or transition function, as this would reduce the role of the policy and make the learning task trivial.

Ultimately, the trade-off between encoding known problem structure through the MDP and learning it from data is a choice that the designer has to make. While standard NAR implicitly enforces the latter, GNARL enables the designer to encode knowledge (if available) through the MDP to the extent that they prefer.

## D.4 BFS / DFS

The BFS algorithm uses the state features in Table 9. Notably, these are the same as for DFS, with the addition of a start node $v_s$ for BFS. The state transition function for both algorithms is given by Algorithm 1. Note that DFS supports directed graphs, while BFS is only run on undirected graphs in the CLRS-30 Benchmark (Veličković et al., 2022). Furthermore, despite these tasks being called *search*, there is no explicit target node that is being searched for, upon which the search can terminate. Instead, the task is to *traverse* the graph and visit all nodes.

We use simple state features as compared to the large number of hints used for DFS in the CLRS-30 Benchmark as they complicate the transition function, whereas we aimed to use the simplest transition function for each environment. The horizon $h = \mathcal{P}(|V| - 1)$. The available actions are $\mathcal{A}(s) = V$ when $p = 1$, and $\mathcal{A}(s) = \{v \mid (\psi_1, v) \in E\}$ when $p = 2$. We do not define an objective function for this problem.

For BFS, a solution is considered correct if the depths of the BFS tree are equivalent to a ground-truth BFS tree. For DFS, there are many possible solutions given the choice of starting node and subsequent node orderings. We use a recursive approach which confirms that each subtree in a spanning forest is valid, described in Algorithm 8 (Nathaniel, 2025).

For BFS, the expert algorithm outputs action distributions in which all edges at the minimum unvisited depth have equal probability of being selected (Algorithm 9). DFS uses a similar approach, but selects the deepest unvisited node instead (Algorithm 10). We collect 1000 episodes of experience from a set of ER graphs with $|V| \in \{4, 7, 11, 13, 16\}$, where $p_{ER}$ is sampled from $[0.1, 0.9]$. We train for 20 epochs, though in practice the BFS model converged after $< 100$ training steps. For validation we use 100 graphs with $|V| = 16$ and $p_{ER} = 0.5$, and we test on graphs with $|V| = 64$ and $p_{ER} = 0.5$.

Table 9: Features for the BFS algorithm.

| Feature | Description | Stage | Location | Type | Initial Value |
|---------|-------------|-------|----------|------|---------------|
| $v_s$ (BFS only) | Start node | Input | Node | Mask One | - |
| adj | Adjacency matrix | Input | Edge | Scalar | - |
| $p$ | Phase ($\mathcal{P} = 2$) | State | Graph | Categorical | 1 |
| $\psi_m$ for $m = 1, \ldots, \mathcal{P}$ | Node last selected in phase $m$ | State | Node | Categorical | $\emptyset$ |
| pred | Predecessor in the tree | State | Node | Pointer | $v \ \forall v \in V$ |
| reach | Node has been searched | State | Node | Mask | $0 \ \forall v \in V$ |

### D.5 Bellman-Ford

The Bellman-Ford algorithm is implemented using the state features in Table 7 and the transition function in Algorithm 3. The non-phase-related state features are equivalent to the hints used in the CLRS-30 Benchmark. The available actions are $\mathcal{A}(s) = \{v \mid \mathtt{mask}_v = 1\}$ when $p = 1$ and $\mathcal{A}(s) = \{v \mid (\psi_1, v) \in E\}$ when $p = 2$. The horizon $h = \mathcal{P}(|V| - 1)|E|$, but a terminal state may be reached when a correct solution pred is found. A solution is considered correct if each shortest path distance matches a reference solution.

If a reward signal is required, a sensible objective function could be $J = \sum_{v \in V} \min(P_{\mathtt{pred}}(v), |V|)$, where $P_{\mathtt{pred}}(v)$ is the length of the path given by following the predecessor of $v$ as per pred until $v_s$ is reached (assuming normalised edge weights).

For the Bellman-Ford expert demonstrations, we use an algorithm that outputs equal probabilities for all edges of a node being expanded (Algorithm 11), reflecting the parallel expansion used in the CLRS-30 Benchmark. Expert experience consists of 10,000 graphs with the same sizes and specifications used for BFS/DFS.

### D.6 MST-Prim

We implement MST-Prim using the features in Table 10 and the transition function in Algorithm 4. We use fewer state features to the hints in the CLRS-30 Benchmark, omitting the u hint. The horizon $h = \mathcal{P}|V|^2$, but a terminal state may be reached when a correct solution pred is found. The available actions are $\mathcal{A}(s) = \psi_1 \cup \{v \mid \mathtt{in\_queue}_v = 1\}$ when $p = 1$ and $\mathcal{A}(s) = \{v|(\psi_1, v) \in E\}$ when $p = 2$. A solution is considered correct if the output is a spanning tree and the total weight of the tree is equal to that of a reference solution.

Expert demonstrations are generated using a Markovian implementation of the MST-Prim algorithm provided by the CLRS-30 Benchmark (Veličković et al., 2022), where all edges satisfying the DP equation are assigned an equal probability (Algorithm 12). Expert experience consists of 10,000 graphs with the same sizes and specifications used for BFS/DFS.

Table 10: Features for the MST-Prim algorithm.

| Feature | Description | Stage | Location | Type | Initial Value |
|---------|-------------|-------|----------|------|---------------|
| $A$ | Weight matrix | Input | Edge | Scalar | - |
| $v_s$ | Start node | Input | Node | Mask One | - |
| $p$ | Phase ($\mathcal{P} = 2$) | State | Graph | Categorical | 1 |
| $\psi_m$ for $m = 1, \ldots, \mathcal{P}$ | Node last selected in phase $m$ | State | Node | Categorical | $\emptyset$ |
| pred | Predecessor in the tree | State | Node | Pointer | $v \ \forall v \in V$ |
| key | Node's key | State | Node | Scalar | $0 \ \forall v \in V$ |
| mark | Node is currently being searched | State | Node | Mask | $0 \ \forall v \in V$ |
| in_queue | Node is in queue | State | Node | Mask | $0 \ \forall v \in V$ |

---

**Algorithm 4:** MST-Prim Transition Function $\mathcal{T}$

---

**1** **Function** STEPSTATE($v$)
**2**   **if** $p = 1$ **then**
**3**     $\texttt{mark}_v \leftarrow 1$
**4**     $\texttt{in\_queue}_v \leftarrow 0$
**5**   **if** $p = 2$ **then**
**6**     $u \leftarrow \psi_1$
**7**     **if** $(u, v) \in E$ and $\texttt{mark}_v = 0$ **then**
**8**       **if** $\texttt{in\_queue}_v = 0$ or $A_{u,v} < \texttt{key}_v$ **then**
**9**         $\texttt{pred}_v \leftarrow u$
**10**         $\texttt{key}_v \leftarrow A_{u,v}$
**11**         $\texttt{in\_queue}_v \leftarrow 1$

**12**   $\psi_p \leftarrow v$
**13**   $p \leftarrow (p \bmod \mathcal{P}) + 1$

---

### D.7 TSP

**Definition 1** (Travelling Salesperson Problem). *Let $K_n = (V, E)$ be a complete graph of $n$ nodes, with edge weights $w_{(u,v)} \in \mathbb{R}_{>0}$ for $u, v \in V$. Let $T$ be a permutation of $V$ called a* tour. *The optimal tour of $K_n$ maximises the objective $J = -\sum_{k=1}^{n-1} w_{(T_k, T_{k+1})} + w_{(T_n, T_1)}$.*

For the TSP, we use the same input features as Georgiev et al. (2024a), with state features listed in Table 11. As there is only one phase, the phase feature is not strictly necessary, but we include it for consistency with the other environments. The horizon $h = |V|$, as each node must be selected once. The available actions are $\mathcal{A}(s) = \{v \mid \texttt{in\_tour}_v = 0\}$. To maintain consistency with Georgiev et al. (2024a), we include a starting node in the input features and require this to be the first node selected without loss of generality, so $\mathcal{A}(s_0) = v_s$.

Training data is composed of the first 10% of the graphs used in Georgiev et al. (2024a) for each of the sizes $|V| \in \{10, 13, 16, 19, 20\}$. We use their full validation and test datasets.

Expert demonstrations are created using tours from the Concorde solver (Applegate et al., 1998), where the first selected node is $v_s$, the next node is the node following $v_s$ in the tour, and so on. BC training is performed on 20 epochs of the $5 \times 10^4$ episodes, and PPO training is performed using $10^7$ steps from episodes on randomly sampled training graphs.

Table 11: Features for the TSP.

| Feature | Description | Stage | Location | Type | Initial Value |
|---------|-------------|-------|----------|------|---------------|
| $A$ | Weight matrix | Input | Edge | Scalar | - |
| $v_s$ | Start node | Input | Node | Mask One | - |
| $\texttt{in\_tour}$ | Node in existing partial tour | State | Node | Mask | $0 \; \forall v \in V$ |
| $p$ | Phase ($\mathcal{P} = 1$) | State | Graph | Categorical | 1 |
| $\psi_m$ for $m = 1$ | Node last selected in phase $m$ | State | Node | Categorical | $\emptyset$ |
| $\texttt{pred}$ | Next node in the tour | State | Node | Pointer | $v \; \forall v \in V$ |

---

**Algorithm 5:** TSP Transition Function $\mathcal{T}$

---

**1** **Function** STEPSTATE($v$)

**2** $\quad$ in_tour$_v \leftarrow 1$

**3** $\quad u \leftarrow \psi_1$

**4** $\quad$ pred$_v \leftarrow$ pred$_u$

**5** $\quad$ pred$_u \leftarrow v$

**6** $\quad \psi_1 \leftarrow v$

---

### D.8 MVC

**Definition 2** (Minimum Vertex Cover). *Given an undirected graph $G = (V, E)$, a* vertex cover *for $G$ is a set $C \subseteq V$ such that $\forall (u, v) \in E$, at least one of $u, v$ is in $C$. Given a node weight $w_v \in \mathbb{R}_{>0}$ for each $v \in V$, a Minimum Vertex Cover is a vertex cover that maximises the objective $J = -\sum_{c \in C} w_c$.*

We model the MVC MDP as a sequential selection of nodes comprising the vertex cover. A solution is valid if each edge has at least one endpoint in the cover. The features used for MVC are found in Table 12, and the transition function is found in Algorithm 6. The horizon $h = |V|$, but a terminal state is entered when the current set of selected nodes forms a valid cover, meaning that most episodes have length $< h$. The available actions are $\mathcal{A}(s) = \{v \mid$ in_cover$_v = 0\}$, preventing selection of nodes that are already in the cover and avoiding the need for beam search or a clean-up stage.

Training data is taken from He & Vitercik (2025), consisting of $10^3$ BA graphs with $M \in [1, 10]$ and $|V| = 16$. We generate $10^4$ episodes of experience from this training set, and train for 10 epochs. Validation and test data is taken from the same source, with 100 graphs of $|V| = 16$ for validation and 100 graphs for each test size.

The expert policy is generated from solutions found using an optimal ILP solver, formulated per He & Vitercik (2025). Each unselected node in the optimal solution is assigned an equal probability of being selected next. For BC, we train using the expert policy distributions. For PPO, we train for $10^7$ steps using episodes on randomly sampled training graphs.

Table 12: Features for MVC.

| Feature | Description | Stage | Location | Type | Initial Value |
|---|---|---|---|---|---|
| adj | Adjacency matrix | Input | Edge | Mask | - |
| $w$ | Node weights | Input | Node | Scalar | - |
| in_cover | Node is in the cover | State | Node | Mask | $0 \ \forall v \in V$ |
| $p$ | Phase ($\mathcal{P} = 1$) | State | Graph | Categorical | 1 |
| $\psi_m$ for $m = 1$ | Node last selected in phase $m$ | State | Node | Categorical | $\emptyset$ |

---

**Algorithm 6:** MVC Transition Function $\mathcal{T}$

---

**1** **Function** STEPSTATE($v$)

**2** $\quad$ in_cover$_v \leftarrow 1$

**3** $\quad \psi_1 \leftarrow v$

---

### D.9 RGC

**Definition 3** (Robust Graph Construction). *Let $G_0 = (V, E_0)$ be a graph and let $\ell < |V|^2 - |E|$ be a positive integer. For $i = 1, \ldots, \ell$, choose a new edge $(u, v) \notin E_{i-1}$, $u, v \in V$, and construct $G_i = (V, E_i)$ where $E_i = E_{i-1} \cup (u, v)$. Define the critical fraction $\xi_G \in (0, 1]$ be the fraction of nodes removed from $G$ in some order until $G$ becomes disconnected. Let $F(G)$ be the expectation of $\xi_G$ under a given removal strategy. Then the objective is to maximise $J = F(G_\ell) - F(G_0)$.*

The features for the RGC environment are found in Table 13, with the transition function shown in Algorithm 7. The horizon corresponds to the edge addition budget $\ell = \lceil \tau(|V|^2 - |V|)/2 \rceil$, where $\tau$, here $\tau = 0.05$, is an input parameter determining the proportion of edges to be added. The available actions are $\mathcal{A}(s) = \{v \mid (u,v) \notin E \text{ for some } u \in V\}$ when $p = 1$ and $\mathcal{A}(s) = \{v \mid (\psi_1, v) \notin E\}$ when $p = 2$.

We show results for both ER graphs with $p = 0.2$ and BA graphs with $M = 2$. All problem instance data is taken from Darvariu et al. (2021a), with training data made up of $10^4$ graphs with $|V| = 20$, validation consisting of 100 graphs with $|V| = 20$, and test data comprising 100 graphs of each size $|V| \in \{20, 40, 60, 80, 100\}$. The weak expert policy is trained using one epoch of demonstrations from the greedy policy for each training graph. The full numerical results corresponding to the RGC evaluation reported in the main text can be found in Table 14.

Table 13: Features for RGC.

| Feature | Description | Stage | Location | Type | Initial Value |
|---|---|---|---|---|---|
| adj | Adjacency matrix of current graph | State | Edge | Mask | - |
| $p$ | Phase ($\mathcal{P} = 2$) | State | Graph | Categorical | 1 |
| $\psi_1$ | Node selected in phase $m$ | State | Node | Categorical | $\emptyset$ |
| $\tau_\ell$ | Remaining edge addition budget (fraction) | State | Graph | Scalar | 1 |

**Algorithm 7:** RGC Transition Function $\mathcal{T}$

1 **Function** STEPSTATE($v$)
2   **if** $p = 2$ **then**
3     $u \leftarrow \psi_1$
4     $\text{adj}_{u,v} \leftarrow 1$
5     $\text{adj}_{v,u} \leftarrow 1$
6     $\tau_\ell \leftarrow \tau_\ell - 2/(|V|^2 - |V|)$
7   $\psi_p \leftarrow v$
8   $p \leftarrow (p \bmod \mathcal{P}) + 1$

Table 14: Full results from the RGC evaluation.

| Graph Type | Objective Function | Method | $\|V\| = 20$ ($\ell = 10$) | $\|V\| = 30$ ($\ell = 22$) | $\|V\| = 40$ ($\ell = 39$) | $\|V\| = 50$ ($\ell = 62$) | $\|V\| = 60$ ($\ell = 89$) | $\|V\| = 70$ ($\ell = 121$) | $\|V\| = 80$ ($\ell = 158$) | $\|V\| = 90$ ($\ell = 201$) | $\|V\| = 100$ ($\ell = 248$) |
|---|---|---|---|---|---|---|---|---|---|---|---|
| BA | random | GNARL$_{\text{PPO}}$ | $0.207_{\pm 0.001}$ | $0.268_{\pm 0.003}$ | $0.318_{\pm 0.006}$ | $0.373_{\pm 0.011}$ | $0.419_{\pm 0.016}$ | $0.445_{\pm 0.018}$ | $0.460_{\pm 0.020}$ | $0.487_{\pm 0.023}$ | $0.518_{\pm 0.027}$ |
| BA | random | GNARL$_{\text{WE}}$ | $0.210_{\pm 0.002}$ | $0.280_{\pm 0.001}$ | $0.341_{\pm 0.002}$ | $0.426_{\pm 0.001}$ | $0.475_{\pm 0.002}$ | $0.498_{\pm 0.001}$ | $0.522_{\pm 0.001}$ | $0.569_{\pm 0.002}$ | $0.602_{\pm 0.003}$ |
| BA | random | GNARL$_{\text{WE+PPO}}$ | $0.209_{\pm 0.001}$ | $0.275_{\pm 0.002}$ | $0.330_{\pm 0.006}$ | $0.398_{\pm 0.010}$ | $0.452_{\pm 0.012}$ | $0.478_{\pm 0.013}$ | $0.497_{\pm 0.015}$ | $0.535_{\pm 0.018}$ | $0.569_{\pm 0.021}$ |
| BA | random | RNet-DQN | $0.213_{\pm 0.001}$ | $0.272_{\pm 0.004}$ | $0.325_{\pm 0.007}$ | $0.396_{\pm 0.012}$ | $0.447_{\pm 0.015}$ | $0.467_{\pm 0.015}$ | $0.487_{\pm 0.016}$ | $0.526_{\pm 0.019}$ | $0.561_{\pm 0.021}$ |
| BA | random | Greedy | $0.133$ | $0.181$ | $0.230$ | $0.271$ | - | - | - | - | - |
| BA | targeted | GNARL$_{\text{PPO}}$ | $0.331_{\pm 0.001}$ | $0.447_{\pm 0.004}$ | $0.505_{\pm 0.012}$ | $0.534_{\pm 0.018}$ | $0.538_{\pm 0.024}$ | $0.547_{\pm 0.027}$ | $0.553_{\pm 0.032}$ | $0.553_{\pm 0.033}$ | $0.558_{\pm 0.034}$ |
| BA | targeted | GNARL$_{\text{WE}}$ | $0.111_{\pm 0.011}$ | $0.051_{\pm 0.007}$ | $0.031_{\pm 0.006}$ | $0.021_{\pm 0.004}$ | $0.015_{\pm 0.002}$ | $0.013_{\pm 0.002}$ | $0.017_{\pm 0.002}$ | $0.017_{\pm 0.002}$ | $0.016_{\pm 0.002}$ |
| BA | targeted | GNARL$_{\text{WE+PPO}}$ | $0.330_{\pm 0.002}$ | $0.443_{\pm 0.004}$ | $0.488_{\pm 0.012}$ | $0.491_{\pm 0.020}$ | $0.481_{\pm 0.024}$ | $0.474_{\pm 0.027}$ | $0.470_{\pm 0.031}$ | $0.466_{\pm 0.032}$ | $0.458_{\pm 0.034}$ |
| BA | targeted | RNet-DQN | $0.272_{\pm 0.002}$ | $0.334_{\pm 0.008}$ | $0.309_{\pm 0.016}$ | $0.283_{\pm 0.022}$ | $0.261_{\pm 0.024}$ | $0.259_{\pm 0.029}$ | $0.259_{\pm 0.032}$ | $0.270_{\pm 0.037}$ | $0.278_{\pm 0.040}$ |
| BA | targeted | Greedy | $0.117$ | $0.083$ | $0.079$ | $0.060$ | - | - | - | - | - |
| ER | random | GNARL$_{\text{PPO}}$ | $0.254_{\pm 0.001}$ | $0.221_{\pm 0.002}$ | $0.152_{\pm 0.003}$ | $0.124_{\pm 0.003}$ | $0.091_{\pm 0.003}$ | $0.078_{\pm 0.003}$ | $0.067_{\pm 0.002}$ | $0.057_{\pm 0.002}$ | $0.051_{\pm 0.002}$ |
| ER | random | GNARL$_{\text{WE}}$ | $0.254_{\pm 0.001}$ | $0.224_{\pm 0.001}$ | $0.161_{\pm 0.001}$ | $0.135_{\pm 0.001}$ | $0.100_{\pm 0.001}$ | $0.085_{\pm 0.001}$ | $0.072_{\pm 0.001}$ | $0.061_{\pm 0.001}$ | $0.054_{\pm 0.001}$ |
| ER | random | GNARL$_{\text{WE+PPO}}$ | $0.256_{\pm 0.001}$ | $0.221_{\pm 0.002}$ | $0.153_{\pm 0.004}$ | $0.126_{\pm 0.004}$ | $0.093_{\pm 0.004}$ | $0.081_{\pm 0.003}$ | $0.069_{\pm 0.003}$ | $0.059_{\pm 0.002}$ | $0.053_{\pm 0.002}$ |
| ER | random | RNet-DQN | $0.239_{\pm 0.002}$ | $0.209_{\pm 0.004}$ | $0.148_{\pm 0.003}$ | $0.119_{\pm 0.003}$ | $0.099_{\pm 0.002}$ | $0.08_{\pm 0.002}$ | $0.068_{\pm 0.001}$ | $0.060_{\pm 0.001}$ | $0.054_{\pm 0.001}$ |
| ER | random | Greedy | $0.206$ | $0.192$ | $0.130$ | $0.106$ | - | - | - | - | - |
| ER | targeted | GNARL$_{\text{PPO}}$ | $0.600_{\pm 0.002}$ | $0.650_{\pm 0.010}$ | $0.368_{\pm 0.027}$ | $0.287_{\pm 0.03}$ | $0.228_{\pm 0.031}$ | $0.206_{\pm 0.032}$ | $0.171_{\pm 0.030}$ | $0.135_{\pm 0.028}$ | $0.114_{\pm 0.026}$ |
| ER | targeted | GNARL$_{\text{WE}}$ | $0.468_{\pm 0.004}$ | $0.497_{\pm 0.021}$ | $0.242_{\pm 0.026}$ | $0.139_{\pm 0.024}$ | $0.092_{\pm 0.027}$ | $0.065_{\pm 0.033}$ | $0.053_{\pm 0.029}$ | $0.035_{\pm 0.029}$ | $0.026_{\pm 0.026}$ |
| ER | targeted | GNARL$_{\text{WE+PPO}}$ | $0.601_{\pm 0.002}$ | $0.623_{\pm 0.014}$ | $0.279_{\pm 0.039}$ | $0.165_{\pm 0.044}$ | $0.097_{\pm 0.040}$ | $0.065_{\pm 0.041}$ | $0.057_{\pm 0.032}$ | $0.048_{\pm 0.027}$ | $0.045_{\pm 0.022}$ |
| ER | targeted | RNet-DQN | $0.455_{\pm 0.003}$ | $0.351_{\pm 0.010}$ | $0.264_{\pm 0.022}$ | $0.240_{\pm 0.025}$ | $0.213_{\pm 0.025}$ | $0.186_{\pm 0.025}$ | $0.158_{\pm 0.022}$ | $0.147_{\pm 0.020}$ | $0.132_{\pm 0.019}$ |
| ER | targeted | Greedy | $0.463$ | $0.579$ | $0.522$ | $0.476$ | - | - | - | - | - |

# E  Training Details

## E.1  Hyperparameter Search

In order to find the best hyperparameters for the model on different problems, a search of the hyperparameter space was conducted within our computational budget. For each model trained with a single method, a grid search was conducted over the values in Table 15. For computational efficiency, we considered only the MPNN processor type which only performs message passing over existing edges in the graph. In runs using PPO for fine-tuning, the policy network and PPO parameters were chosen from the best PPO run, and the BC parameters were then selected from the highest-scoring BC run with the same policy network parameters.

Table 15: Hyperparameter values used in grid search.

| Property | Values searched |
|---|---:|
| *Policy Network* | |
| Processor type | MPNN |
| Pooling type | Max, Mean |
| $L$ | 1, 2, 3, 4 |
| MLP layers | 2, 3 |
| Aggregation | Max, Sum |
| *Behavioural Cloning* | |
| Learning rate | 5.0e-2, 1.0e-3, 5.0e-4, 1.0e-4 |
| Batch size | 8, 16, 32, 64, 128 |
| *PPO* | |
| Learning rate | 5.0e-4, 1.0e-5, 5.0e-6 |
| Batch size | 32, 64, 128 |

We test both Max and Sum aggregation functions in the MPNN. Ibarz et al. (2022) suggest that the Max aggregation provides good algorithmic alignment for the CLRS-30 Benchmark domains. However, this causes state aliasing for certain domains where the node and edge features are identical in the initial state, such as RGC, so we also consider Sum aggregation in the hyperparameter search.

The final model for evaluation was chosen based on the best achieved validation score, corresponding to mean success for CLRS-30 Benchmark domains and mean reward for NP-hard domains. For runs using PPO, this score was taken over the average of 5 different seeds. In case of ties (for example when multiple models achieved mean success of 1), the tie was broken using the mean number of steps. For BFS and DFS, the number of steps is fixed, so the final hyperparameters were chosen to be the individual parameters which were most common among runs with mean success of 1. The final hyperparameter choices can be found in Table 16.

For all experiments, we used $\gamma = 1$. For PPO we used an update interval of 1024 steps, 10 epochs. Other PPO hyperparameters were set according to the defaults in Stable Baselines 3 (Raffin et al., 2021). For BC we ran evaluation every 100 batches. For PPO we ran evaluation at every update. The best model was chosen according to a lexicographical ordering of success rate, mean reward, and negative mean episode length.

## E.2  Comparison of PPO and BC Training

Figure 7 shows the validation set performance of GNARL during training as a function of training time for both BC and PPO on the TSP problem. Here, PPO was trained for $10^7$ episode steps, while BC was trained on 50,000 episodes for 20 epochs ($10^6$ episode steps). PPO reaches a higher performance on the validation set more quickly than BC, but we also see from results in Tables 4 and 20 that policies trained using BC appear to generalise better to larger graphs than those trained using PPO. The investigation of this phenomenon is left to future work.

Table 16: Chosen hyperparameter values for each domain.

| Experiment | Aggr | Pooling | L | MLP layers | LR (BC) | Batch (BC) | LR (PPO) | Batch (PPO) |
|---|---|---|---|---|---|---|---|---|
| | | | *CLRS* | | | | | |
| BFS | Max | Mean | 1 | 2 | 1.0e-3 | 16 | - | - |
| DFS | Max | Max | 2 | 2 | 5.0e-2 | 8 | - | - |
| Bellman-Ford | Max | Mean | 2 | 2 | 5.0e-4 | 8 | - | - |
| MST-Prim | Max | Max | 4 | 3 | 1.0e-3 | 16 | - | - |
| | | | *TSP* | | | | | |
| BC | Max | Max | 4 | 2 | 1.0e-3 | 32 | - | - |
| PPO | Max | Max | 4 | 2 | - | - | 5.0e-4 | 128 |
| WE + PPO | Max | Max | 4 | 2 | 5.0e-4 | 8 | 5.0e-4 | 128 |
| | | | *MVC* | | | | | |
| BC | Sum | Max | 4 | 2 | 1.0e-3 | 32 | - | - |
| PPO | Sum | Mean | 4 | 2 | - | - | 1.0e-5 | 64 |
| | | | *RGC* | | | | | |
| PPO (BA-R) | Sum | Mean | 4 | 3 | - | - | 5.0e-4 | 32 |
| PPO (ER-R) | Sum | Mean | 2 | 2 | - | - | 5.0e-4 | 32 |
| WE + PPO (BA-R) | Sum | Mean | 4 | 3 | 1.0e-4 | 8 | 5.0e-4 | 32 |
| WE + PPO (ER-R) | Sum | Mean | 2 | 2 | 1.0e-3 | 8 | 5.0e-4 | 32 |
| PPO (BA-T) | Sum | Max | 4 | 3 | - | - | 5.0e-4 | 128 |
| PPO (ER-T) | Sum | Max | 4 | 2 | - | - | 5.0e-4 | 32 |
| WE + PPO (BA-T) | Sum | Max | 4 | 3 | 5.0e-4 | 8 | 5.0e-4 | 128 |
| WE + PPO (ER-T) | Sum | Max | 4 | 2 | 1.0e-3 | 16 | 5.0e-4 | 32 |

### E.3 Training Time

We provide training times for GNARL on the CLRS-30 Benchmark algorithms in Table 17. Due to the difference in implementation frameworks (PyTorch Geometric and Gymnasium for GNARL, and JAX for NAR), we cannot provide a direct comparison to the TripletMPNN model (Ibarz et al., 2022). For GNARL, training is performed over a given number of episodes, meaning that the number of steps taken in training depends on the average length of the expert's trajectory (see Appendix C.4). Training was performed on a single core of an Intel Platinum 8628 CPU with 4GB of memory using CentOS Linux 8.1. For BFS and DFS, GNARL uses early stopping when a perfect solution is found on the training set, which reduces training time. As Bellman-Ford and MST-Prim have variable-length episodes, training continues after a model achieves 100% validation performance, as the model may still improve its efficiency in finding solutions.

Table 17: Training time (hours) for GNARL models on CLRS-30 Benchmark algorithms. BFS and DFS use early stopping.

| BFS | DFS | Bellman-Ford | MST-Prim |
|---|---|---|---|
| $0.04_{\pm 0.03}$ | $0.02_{\pm 0.01}$ | $116_{\pm 1}$ | $197_{\pm 6}$ |

### E.4 Effect of Critic Network on Training Time

When training GNARL using imitation learning, it is not necessary to train the critic network unless the module will later be fine-tuned using an actor-critic method such as PPO. For the TSP problem, the GNARL model trained using BC for 5000 episodes with the critic enabled took $27.4_{\pm 1.3}$ hours, while training without

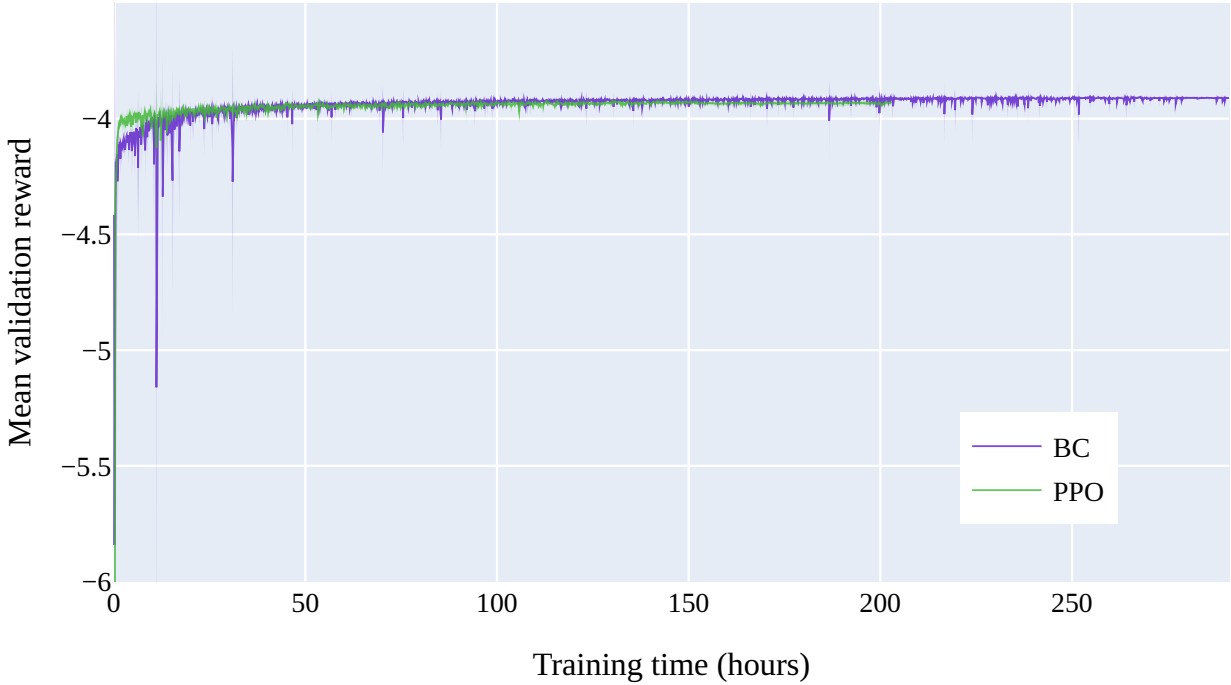

Figure 7: Comparison of BC and PPO training on TSP, averaged over 5 seeds.

the critic took $20.1_{\pm 1.1}$ hours (averaged over 5 seeds). The results demonstrate that training the critic network increases training time by approximately 36%.

## F    Evaluation Details

### F.1    CLRS-30 Inference Time

We provide inference time measurements for GNARL, as evaluated on the CLRS-30 Benchmark, in Table 18. A direct comparison of inference times between GNARL and NAR models on the CLRS-30 Benchmark algorithms is not possible due to significant differences in implementation frameworks. GNARL is implemented in PyTorch Geometric (Fey & Lenssen, 2019) and Gymnasium (Towers et al., 2024), while NAR models are implemented in JAX (Bradbury et al., 2018), providing speed-ups of several orders of magnitude through function compilation. The times were measured on a single core of an Intel Platinum 8628 CPU, and are the average over 5 models on the test set of 100 graphs. The total time includes the entire inference process including both model forward pass and environment step, while the environment time reports only the time spent in the environment step function.

One factor influencing inference time is the level of parallelism built into the representation of the algorithm. A more parallel representation leads to shorter episode lengths, reducing both the number of inference steps and the number of environment steps required during evaluation. This design choice is further discussed in Appendix C.4. Additionally, the inference time of GNARL is influenced by the success rate of the model for certain problems where successful episodes terminate earlier, reducing the average episode length. The number of steps in an unsuccessful episode in MST-Prim is approximately $18\times$ longer than a successful episode, leading to a large increase in inference time due to the low average success rate. This effect is also present in Bellman-Ford, though to a lesser extent due to the lower horizon of the environment and higher success rate of the models.

Table 18: Inference time (seconds per graph) for GNARL models on graphs with $|V| = 64$.

| Runs | Inference | Environment | Total |
|------|-----------|-------------|-------|
| *BFS* | | | |
| All | $0.936_{\pm0.091}$ | $0.167_{\pm0.004}$ | $1.10_{\pm0.10}$ |
| *DFS* | | | |
| All | $1.50_{\pm0.01}$ | $0.134_{\pm0.007}$ | $1.64_{\pm0.00}$ |
| *Bellman-Ford* | | | |
| All | $5.46_{\pm0.35}$ | $2.84_{\pm0.12}$ | $8.30_{\pm0.40}$ |
| Success Only | $4.94_{\pm0.31}$ | $2.78_{\pm0.15}$ | $7.73_{\pm0.46}$ |
| Failure Only | $11.5_{\pm1.0}$ | $3.23_{\pm0.42}$ | $14.7_{\pm1.0}$ |
| *MST-Prim* | | | |
| All | $108_{\pm24}$ | $2.41_{\pm0.49}$ | $110_{\pm24}$ |
| Success Only | $13.7_{\pm0.1}$ | $0.378_{\pm0.013}$ | $14.1_{\pm0.1}$ |
| Failure Only | $243_{\pm2}$ | $5.34_{\pm0.09}$ | $249_{\pm2}$ |

## G  Additional Experiments

### G.1  Graph Accuracy and Solution Correctness for NAR Models

Given a single node ordering, the graph accuracy of a model is calculated as the proportion of graphs for which all output predictions match the expected solution. Conversely, solution correctness is calculated as the proportion of graphs for which the predicted solution could be produced by the reference algorithm under any node ordering. If an algorithm has a unique solution for each input graph, then graph accuracy and solution correctness are equivalent. However, when multiple solutions are possible for the algorithm, graph accuracy is a lower bound for solution correctness, as it is possible that a model could predict a correct solution which does not strictly correspond to the expected output label.

For Bellman-Ford and MST-Prim, edge weights are randomly sampled from a uniform distribution, meaning the probability of multiple solutions existing is 0, so graph accuracy and solution correctness are equivalent. However, for BFS and DFS, multiple solutions are possible for a given input graph. In BFS, any tree with root $v_s$ and with all nodes at the correct depth from $v_s$ is a correct solution. In DFS, any valid depth-first traversal of the graph is a correct solution, with no restrictions on the root node, meaning many solutions exist.

We evaluate the graph accuracy and solution correctness of models for BFS and DFS produced by the TripletMPNN trained per Ibarz et al. (2022) in Table 19, with the evaluation run over 100 graphs with $|V| = 64$. The results confirm that graph accuracy is not directly equivalent to solution correctness for algorithms with many possible solutions. We also include the estimated graph accuracy based on the node accuracy (micro-$F_1$ score) alone.

Table 19: Graph accuracy and solution correctness (%) for TripletMPNN over 5 seeds.

| | Solution correctness (any ordering) | Graph accuracy (single ordering) | Estimated graph accuracy (micro-$F_1^{|V|}$) |
|------|-------------------------------------|----------------------------------|------------------------------------------------|
| BFS | $100.0_{\pm0.0}$ | $87.4_{\pm10.1}$ | $85.0_{\pm12.7}$ |
| DFS | $24.2_{\pm28.6}$ | $0.0_{\pm0.0}$ | $0.0_{\pm0.0}$ |
| Bellman-Ford | $12.4_{\pm6.5}$ | $12.4_{\pm6.5}$ | $14.2_{\pm5.3}$ |
| MST-Prim | $2.6_{\pm4.2}$ | $2.6_{\pm4.2}$ | $0.6_{\pm0.9}$ |

From the empirical evaluation, we observe that the approximation for graph accuracy based on independent node accuracies is reasonable, as it falls within 3 percentage points of the true graph accuracy for each domain. For BFS and DFS, the impact of multiple possible solutions is much greater, with a difference of 24.2 percentage points between solution correctness and graph accuracy for DFS. The approximation does not impact the trend of the results presented in Table 2, as most methods either performed very well or very poorly on BFS and DFS, with GNARL$_{BC}$ achieving 100% on both. For Bellman-Ford and MST-Prim, GNARL$_{BC}$ greatly outperforms all baselines except DNAR, even allowing extra margin for the approximated graph accuracy.

### G.2 DNAR on Combinatorial Optimisation Problems

The DNAR approach from Rodionov & Prokhorenkova (2025) is optimised for discrete processing in algorithmic environments, such as the CLRS-30 Benchmark, enabling perfect generalisation on selected problem instances. The architecture utilises codebook embeddings alongside an algorithmically-inspired module which updates the non-scalar features of the environment, followed by a scalar update module which performs operations on the scalar features to match the next hint.

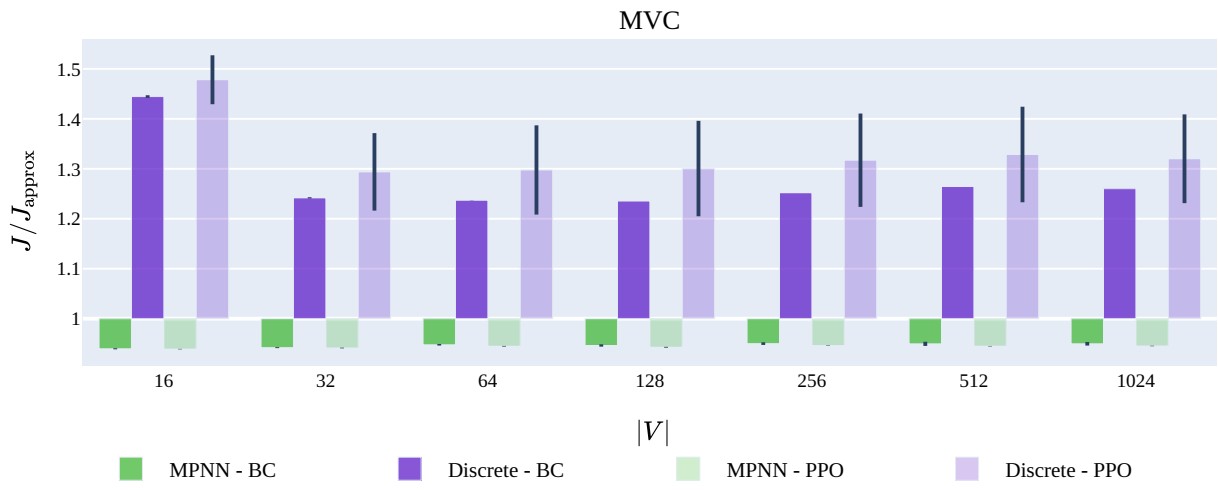

Figure 8: Performance of the GNARL architecture using the discrete processor from Rodionov & Prokhorenkova (2025) on MVC (↓), compared to the MPNN processor.

In order to assess how readily this architecture translates to combinatorial optimisation problems, we replace the MPNN processor of the GNARL architecture with the processor of the DNAR architecture, and train the model on the MVC environment using both BC and PPO. Since the environment updates the features based on the action taken, rather than the model needing to update the scalar features, we do not make use of the scalar update module. Figure 8 shows the results of this evaluation compared to the baseline GNARL results using the MPNN processor. The discrete model clearly struggles to learn to solve the MVC problem, consistently achieving performance much worse than the approximation algorithm. As the MVC environment is relatively straightforward, the results suggest that the discrete processor approach of Rodionov & Prokhorenkova (2025) is not as well-suited to combinatorial optimisation problems as it is for P-hard algorithmic learning.

### G.3 TSP with Weak Expert Warm-Starting

In some circumstances, it is not possible to allocate a large computational budget to PPO training. In such cases, it can be beneficial to pre-train the policy using expert demonstrations, even if only a weak expert is available. We simulate such a scenario for the TSP in which the weak expert (WE) policy greedily selects the

next node to maximise the objective function. In Table 20, we compare the results of training GNARL using PPO for $10^6$ steps against GNARL trained using only the weak expert for $10^5$ episodes (WE), and GNARL trained using the weak expert for $10^5$ episodes followed by PPO fine-tuning for $10^6$ steps (WE+PPO). We also include results for GNARL trained using the weak expert for $10^5$ episodes, followed by PPO fine-tuning for $10^6$ steps using an augmented validation set (WE+PPO+Val). This validation set contains the original validation set with $|V| = 20$ and adds 10 graphs of each size $|V| \in \{40, 60, 80\}$. The results do not demonstrate improvement when using warm-starting with the weak expert, and in fact show a degradation in performance compared to training using only PPO. However, when using the augmented validation set to select the best model for OOD performance, we see an improvement in performance at larger scales. Interestingly, the model trained using only the weak expert is able to outperform Georgiev et al. (2024a) at OOD scales above $4\times$, demonstrating the scalability benefits of the GNARL architecture.

Table 20: Percentage above the optimal baseline on the TSP for models trained using demonstrations from a weak (greedy) expert, with further fine-tuning via PPO.

| | Test size | | | | | |
|---|---|---|---|---|---|---|
| **Model** | 40 (2×) | 60 (3×) | 80 (4×) | 100 (5×) | 200 (10×) | 1000 (50×) |
| GNARL$_{\text{PPO}}$ | 5.0±0.6 | 7.9±1.0 | 10.8±2.4 | 11.8±2.1 | 19.2±2.2 | 31.8±3.6 |
| GNARL$_{\text{WE}}$ | 16.3±4.7 | 18.6±5.0 | 19.3±6.0 | 20.3±6.0 | 22.7±6.3 | 26.7±6.7 |
| GNARL$_{\text{WE + PPO}}$ | 5.3±0.5 | 9.3±1.0 | 12.0±1.5 | 14.0±2.3 | 24.7±4.2 | 63.7±19.5 |
| GNARL$_{\text{WE + PPO+Val}}$ | 4.8±0.1 | 8.0±0.6 | 10.4±0.5 | 12.1±0.9 | 20.2±3.8 | 41.1±20.7 |

### G.4 Effect of Episode Termination on MVC Evaluation

For the MVC environment, we use the objective value of the solution generated by an approximation algorithm (Khuller et al., 1994) as the reference by which other models are normalised. The objective value of this algorithm is given by $J_{\text{approx}} = -\sum_{v \in C_{\text{approx}}} w_v$, where $C_{\text{approx}}$ is the cover produced by the approximation algorithm.

In the design of the MVC MDP, the episode terminates as soon as a valid cover is found. Since the approximation algorithm may include extraneous nodes in the cover $C_{\text{approx}}$, it is possible that a valid cover $C \subset C_{\text{approx}}$ exists. Thus, a model which exactly replicates the covers produced by the approximation algorithm may, by chance, select nodes in such an order that $C$ is found and the MDP terminates before $C_{\text{approx}}$ is selected, achieving a better objective function value.

We evaluate the effect of this truncation at different graph sizes in Table 21. Here, the approximation algorithm is evaluated, with nodes in $C_{\text{approx}}$ chosen with uniform probability, and normalised against the full cover objective $J_{\text{approx}}$. The overall effect is small.

Table 21: Objective ratio for MVC under the expert policy given by the approximation algorithm.

| | Test size | | | | | | |
|---|---|---|---|---|---|---|---|
| | 16 | 32 | 64 | 128 | 256 | 512 | 1024 |
| $J/J_{\text{approx}}$ | 0.9968 | 0.9938 | 0.9976 | 0.9991 | 0.9994 | 0.9996 | 0.9999 |

## H Algorithms

Algorithm 8 shows pseudocode for checking if a predecessor forest is a valid DFS solution for a given graph.

---

**Algorithm 8:** Check if a Predecessor Forest is a Valid DFS Solution

---

**Input:** Adjacency matrix `adj`, predecessor array `pred`
**Output:** True if `pred` is a valid DFS forest for `adj`, else False

**1 Function** CheckValidDFS(*adj*, *pred*)**:**
**2**     Obtain $G = (V, E)$ from `adj`
**3**     `active_nodes` $\leftarrow V$
**4**     **return** IsValidForestRecursive(*active_nodes*, *pred*)

**5 Function** IsValidForestRecursive(*active_nodes*, *pred*)**:**
**6**     **if** $|$`active_nodes`$| \leq 1$ **then**
**7**        **return** True

**8**     `subroots` $\leftarrow \{v \in$ `active_nodes` $\mid$ `pred`$_v \notin$ `active_nodes` or `pred`$_v = v\}$
**9**     **if** `subroots` $= \emptyset$ and `active_nodes` $\neq \emptyset$ **then**
**10**       **return** False

**11**     **if** $|$`subroots`$| > 1$ **then**
**12**       **foreach** $v \in$ `active_nodes` **do**
**13**          subroot$_v \leftarrow i$, where $i$ is the root of the subtree containing $v$ by following `pred`

**14**       $G_{\text{component}} \leftarrow (\text{subroots}, \emptyset)$
**15**       **foreach** $(u, v) \in E$ such that $u, v \in$ `active_nodes` **do**
**16**          **if** subroot$_u \neq$ subroot$_v$ **then**
**17**             $G_{\text{component}}.E \leftarrow G_{\text{component}}.E \cup \{(\text{subroot}_u, \text{subroot}_v)\}$

**18**       **if** $G_{\text{component}}$ has a cycle **then**
**19**          **return** False

**20**     **foreach** `root` in `subroots` **do**
**21**       `descendants` $\leftarrow$ descendants of `root` in `active_nodes`
**22**       **if** `descendants` $\neq \emptyset$ **then**
**23**          **if** not IsValidForestRecursive(*descendants*) **then**
**24**             **return** False

**25**     **return** True

---

## H.1 Expert Policies

In this section, we provide pseudocode for the expert policies used to generate action distributions for the CLRS-30 Benchmark algorithms studied. Each algorithm operates on the current state $s$, which includes the graph $G = (V, E)$ and all node and edge features. The BFS, DFS, Bellman-Ford, and MST-Prim expert algorithms are provided in Algorithms 9–12.

While we implement expert policies that provide the action distribution, it is also possible to implement expert policies that provide a single action demonstration for each state. This is simpler to implement, but requires more training episodes to learn the distribution.

---

**Algorithm 9:** Expert Policy for BFS Environment

---

**1** **Function** $\pi^*(s)$:
**2**    **if** $p = 1$ **then**
**3**      └ **return** PhaseOnePolicy($s$)
**4**    **else**
**5**      └ **return** PhaseTwoPolicy($s, \psi_1$)

**6** **Function** PhaseOnePolicy($s$):
**7**    closed $\leftarrow \{v \in V \mid \texttt{reach}_v = 1 \wedge \texttt{reach}_u = 1 \; \forall u \in \mathcal{N}(v)\}$
**8**    open $\leftarrow V \setminus$ closed
**9**    depths $\leftarrow$ GetDepthCounter(*reach*, *pred*)
**10**    $d_{\min} \leftarrow \min_{v \in \texttt{open}} \texttt{depths}_v$
**11**    eligible $\leftarrow \{j \in \texttt{open} \mid \texttt{depths}_j = d_{\min}\}$
**12**    **return** $\pi(v) = \begin{cases} 1/|\texttt{eligible}| & \text{if } v \in \texttt{eligible} \\ 0 & \text{otherwise} \end{cases}$

**13** **Function** PhaseTwoPolicy($s, \psi_1$):
**14**    $N \leftarrow \{j \in \mathcal{N}(\psi_1) \setminus \{\psi_1\} \mid \texttt{reach}_j = 0\}$
**15**    **return** $\pi(v) = \begin{cases} 1/|N| & \text{if } v \in N \\ 0 & \text{otherwise} \end{cases}$

---

**Algorithm 10:** Expert Policy for DFS Environment

---

**1** **Function** $\pi^*(s)$:
**2**    **if** $p = 1$ **then**
**3**      └ **return** PhaseOnePolicy($s$)
**4**    **else**
**5**      └ **return** PhaseTwoPolicy($s, \psi_1$)

**6** **Function** GetNodeColour(*reach*, *adj*):
**7**    colour $\leftarrow \begin{cases} 0 & \text{if } \texttt{reach}_v = 0 \\ 2 & \text{if } \texttt{reach}_v = 1 \wedge \forall u \in \mathcal{N}(v), \; \texttt{reach}_u = 1 \\ 1 & \text{otherwise} \end{cases}$
**8**    **return** colour

**9** **Function** PhaseOnePolicy($s$):
**10**    colour $\leftarrow$ GetNodeColour(*reach*, *adj*)
**11**    depths $\leftarrow$ GetDepthCounter(*colour* $\neq 0$, *pred*)
**12**    $d_{\max} \leftarrow \max_{v \mid \texttt{colour}_v \neq 2} \texttt{depths}_v$
**13**    eligible $\leftarrow \{v \mid \texttt{colour}_v = 1 \wedge \texttt{depths}_v = d_{\max}\}$
**14**    **if** eligible $= \emptyset$ **then**
**15**      └ eligible $\leftarrow \{v \mid \texttt{colour}_v = 0 \wedge \texttt{depths}_v = d_{\max}\}$
**16**    **return** $\pi(v) = \begin{cases} 1/|\texttt{eligible}| & \text{if } v \in \texttt{eligible} \\ 0 & \text{otherwise} \end{cases}$

**17** **Function** PhaseTwoPolicy($s, \psi_1$):
**18**    $N \leftarrow \{j \in \mathcal{N}(\psi_1) \setminus \{\psi_1\} \mid \texttt{reach}_j = 0\}$
**19**    **if** $N = \emptyset$ **and** $\texttt{reach}_{\psi_1} = 0$ **then**
**20**      └ $N \leftarrow \{\psi_1\}$
**21**    **return** $\pi(v) = \begin{cases} 1/|N| & \text{if } v \in N \\ 0 & \text{otherwise} \end{cases}$

---

---

**Algorithm 11:** Expert Policy for Bellman-Ford Environment

---

**1** **Function** $\pi^*(s)$**:**

**2**    **if** $p = 1$ **then**

**3**      **return** PhaseOnePolicy($s$)

**4**    **else**

**5**      **return** PhaseTwoPolicy($s, \psi_1$)

**6** **Function** PhaseOnePolicy($s$)**:**

**7**    $\texttt{possible} \leftarrow \begin{cases} \emptyset & \text{if } \exists v \in V, \texttt{mask}_v = 1 \\ \{v_s\} & \text{otherwise} \end{cases}$

**8**    **foreach** $v \in V$ where $\texttt{mask}_v = 1$ **do**

**9**      **if** PhaseTwoPolicy($s, v$) is not $\emptyset$ **then**

**10**        $\texttt{possible} \leftarrow \texttt{possible} \cup \{v\}$

**11**    **return** $\pi(v) = \begin{cases} 1/|\texttt{possible}| & \text{if } v \in \texttt{possible} \\ 0 & \text{otherwise} \end{cases}$

**12** **Function** PhaseTwoPolicy($s, u$)**:**

**13**    $N \leftarrow \{v \in \mathcal{N}(u) \setminus \{u\} \mid \texttt{dist}_u + A_{uv} < \texttt{dist}_v \vee \texttt{mask}_v = 0\}$

**14**    **if** $N = \emptyset$ **then**

**15**      **return** $\emptyset$

**16**    **return** $\pi(v) = \begin{cases} 1/|N| & \text{if } v \in N \\ 0 & \text{otherwise} \end{cases}$

---

**Algorithm 12:** Expert Policy for MST-Prim Environment

---

**1** **Function** $\pi^*(s)$**:**

**2**    **if** $p = 1$ **then**

**3**      **return** PhaseOnePolicy($s$)

**4**    **else**

**5**      **return** PhaseTwoPolicy($s, \psi_1$)

**6** **Function** PhaseOnePolicy($s$)**:**

**7**    **if** $\psi_1$ is set and PhaseTwoPolicy($s, \psi_1$) $\neq \emptyset$ **then**

**8**      $k \leftarrow \psi_1$             // Maintain current node if possible

**9**    **else**

**10**      $C \leftarrow \{v \in V \mid \texttt{in\_queue}_v = 1\}$, sorted by increasing $\texttt{key}$

**11**      **foreach** $u \in C$ **do**

**12**        **if** PhaseTwoPolicy($s, u$) $\neq \emptyset$ **then**

**13**          $k \leftarrow u$

**14**          **break**

**15**    **return** $\pi(v) = \begin{cases} 1 & v = u \\ 0 & \text{otherwise} \end{cases}$

**16** **Function** PhaseTwoPolicy($s, \psi_1$)**:**

**17**    $N \leftarrow \{v \in \mathcal{N}(\psi_1) \setminus \{\psi_1\} \mid \texttt{mark}_v = 0 \wedge (\texttt{key}_v > A_{\psi_1 v} \text{ or } \texttt{in\_queue}_v = 0)\}$

**18**    **if** $N = \emptyset$ **then**

**19**      **return** $\emptyset$

**20**    **return** $\pi(v) = \begin{cases} 1/|N| & v \in N \\ 0 & \text{otherwise} \end{cases}$

---

