# OpenReview forum: "Tackling GNARLy Problems: Graph Neural Algorithmic Reasoning Reimagined through Reinforcement Learning"
_TMLR — Under review for TMLR_

### Review · Reviewer_tfmH · 2026-06-26

**Summary Of Contributions:**

## Summary

* Task: Learning to execute graph algorithms with neural networks.
* The paper proposes GNARL, a framework that recasts neural algorithmic reasoning over graph algorithms as an MDP.
* Its contribution is to reinterpret NAR tasks as MDPs and show that a unified graph-policy framework can handle both CLRS-style algorithm execution and several combinatorial optimization tasks.
* The paper evaluates GNARL on CLRS-style graph algorithms, MVC, TSP, and robust graph construction.

## Strengths

* \[S1\] The paper identifies a meaningful limitation of standard NAR.
* \[S2\] The proposed GNARL framework provides a clean conceptual bridge between NAR and graph RL/NCO.
* \[S3\] The paper is generally well organized and includes substantial appendix material.

## Weaknesses

* \[W1\] My biggest concern is that the paper motivates GNARL as “NAR reimagined through RL,” but from the NCO perspective many tasks considered here are already naturally sequential decision problems. The paper should better explain what additional insight is gained by passing through the NAR framing, beyond applying standard graph RL/IL tools to structured graph-output problems.
* \[W2\] Even accepting the NAR-to-RL reframing, the paper should more carefully separate what is learned by the neural policy from what is already encoded in the hand-designed MDP, because the hand-designed MDPs encode a substantial amount of algorithmic reasoning. In several environments, important algorithmic logic is placed in the transition function, action masks, or termination rule. This makes it unclear whether GNARL learns algorithmic reasoning or primarily learns action selection within a manually engineered solver.
* \[W3\] Several claims in the paper need to be scoped or clarified more carefully.
  * The claim that GNARL constructs valid solutions “by construction” is too broad. Action masking prevents certain locally invalid choices, such as selecting a nonexistent edge or revisiting a TSP node, but it does not guarantee global algorithmic correctness. For example, Bellman-Ford and MST-Prim still have success rates below 100%, showing that locally valid action sequences can produce incorrect global outputs.
  * The termination rules for Bellman-Ford and MST-Prim are also unclear. The appendix states that episodes may terminate when a correct solution is reached, but correctness is defined by comparison with a reference solution. If reference-solution information is used for early stopping during inference, then the reported step counts and inference times may not reflect a deployable procedure.
  * The CLRS comparisons also mix measured solution correctness with estimated graph accuracy based on node accuracy. Estimating graph accuracy as node\_accuracy^|V| assumes independence and may not reliably reflect true graph-level correctness.
* \[W4\] A central motivation of GNARL is that, unlike standard supervised NAR, it can be trained without an expert algorithm using only rewards. The experiments show some promising cases, but the overall evidence is more mixed than the conclusion suggests.
  * PPO performs poorly or unstably on Bellman-Ford and is sensitive to reward design on CLRS tasks. TSP-PPO generalizes poorly to the largest test instances unless additional heuristic help is introduced.
  * The RGC results are presented mainly as plots without detailed numerical tables or uncertainty estimates, making it difficult to judge robustness.

**Audience:**

Yes

**Audience Explanation:**

The paper would likely interest at least a meaningful subset of the TMLR audience, especially researchers working on neural algorithmic reasoning, graph neural networks, reinforcement learning for combinatorial optimization, and neural combinatorial optimization.

**Broader Impact Concerns:**

I have no specific broader-impact concerns that require further discussion.

**Claims And Evidence:**

No

**Claims Explanation:**

See also \[Weaknesses\] above. Specifically,

* First, the paper should better distinguish what is learned by the neural policy from what is encoded in the hand-designed MDP. Across tasks, the environment specifies state variables, action masks, transition functions, horizons, rewards, and termination rules. In several cases, substantial algorithmic logic is placed in these components, e.g., edge relaxation in Bellman-Ford, key/predecessor updates in MST-Prim, partial-tour maintenance in TSP, and cover-validity checking in MVC. This makes it difficult to assess whether GNARL learns algorithmic reasoning or mainly learns action selection within a manually engineered solver.
* Second, some claims are broader than the evidence supports. In particular, the claim that GNARL constructs valid solutions “by construction” should be scoped more carefully. Action masking prevents locally invalid choices, but it does not guarantee global algorithmic correctness for tasks such as Bellman-Ford or MST-Prim, where the reported success rates remain below 100%.
* Third, some evaluation details are ambiguous. The appendix states that Bellman-Ford and MST-Prim episodes may terminate when a correct solution is reached, but correctness is defined by comparison with a reference solution. If such reference information is used for early stopping during inference, then reported episode lengths and inference times may not reflect a deployable procedure.
* Finally, the evidence for expert-free RL is mixed. PPO performs well on MVC and appears useful in some RGC settings, but is unstable or weak on Bellman-Ford, sensitive to reward design on CLRS tasks, and generalizes poorly to the largest TSP instances without additional heuristic help. Thus, the experiments support a more limited claim: GNARL can work well in some settings, but the broader claims about general expert-free applicability and validity-by-construction are not yet fully supported.

**Requested Changes:**

Related to \[Weakness\] above (Critical for acceptance):

* Related to \[W1\]. The paper should more explicitly state that MDP-based graph solution construction, GNN policies, action masking, behavioral cloning, and PPO are standard tools in neural combinatorial optimization and graph RL. The claimed novelty should be narrowed to the NAR-specific reframing: converting NAR-style algorithm traces/hints into MDP state/action formulations and demonstrating this bridge across several graph algorithmic tasks.
* Related to \[W2\]. For each evaluated task, the authors should explicitly identify which parts of the algorithmic logic are implemented in the environment and which decisions are learned by the neural policy. This should cover state features, action masks, transition functions, rewards, horizons, and termination rules.
* Related to \[W3\]. The paper should distinguish local action validity, feasible solution construction, global algorithmic correctness, and objective optimality. Action masks can prevent locally invalid actions, but they do not guarantee correct Bellman-Ford or MST-Prim outputs.
* Related to \[W3\]. The appendix states that episodes may terminate when a correct solution is reached, while correctness is defined by comparison to a reference solution. The authors should specify exactly what information is used for termination during validation/test inference. If reference-solution information is used, they should rerun or additionally report results under a deployable stopping rule, such as a fixed horizon or an internally checkable convergence condition.
* Related to \[W3\]. Table 2 mixes measured solution correctness with estimated graph accuracy derived from node accuracy. The authors should either measure graph-level solution correctness for the baselines under the same evaluation protocol, clearly mark these entries as non-comparable estimates, or avoid using them to support strong comparative claims.
* Related to \[W4\]. The authors should revise the conclusion and abstract-level claims to reflect that reward-only training works well in some settings but is unstable or weak in others. Alternatively, they could provide a more systematic analysis of when PPO succeeds or fails, including reward sensitivity, variance across seeds, and exact numerical results for RGC.

I also have some additional points (Would strengthen the work):

* It would be useful to add ablations isolating the contribution of key design choices.
* It would be useful to provide exact numerical RGC results with uncertainty. The RGC experiment is important because it supports the no-expert setting, but the current plot-based presentation makes robustness difficult to assess.
* It would be useful to discuss computational tradeoffs more prominently. GNARL often requires many more sequential steps than standard NAR and has large training times for some CLRS tasks.

---

> ### Author Response · Authors · 2026-07-21
>
> Many thanks for your detailed review and the opportunity to sharpen the scope of our claims, discuss important design trade-offs, and clarify some of the experimental details. Please see our responses below.
>
> ### W1 - Novelty claims
>
> We fully concur that the tools listed by the reviewer (MDP-based graph solution construction, GNN policies, etc.) are standard in the NCO and Graph RL literatures. We did not claim novelty of the tools relative to these fields. Indeed, the key novelty of our work is the NAR-specific reframing: demonstrating that the NAR principle of learning an algorithm using traces and hints can be explicitly aligned with trajectories and states / actions in the MDP formalism; rendering powerful IL and RL tools applicable to NAR.
>
> From the point of view of NCO, we consider that the main insight of the NAR framing is the highly reusable framework. While the NCO literature generally hand-designs pipelines for a single problem or class of problems, GNARL merely requires specifying the state variables and transition function in order to be applied to a new NCO problem. We anticipate that this is most useful when solving an under-studied CO problem such as RGC, for which GNARL allows building an initial learning-based solver with minimal effort.
>
> **Changes made:** We have added an explicit statement in the first paragraph of Section 4 to make the scope of claims clearer.
>
> ### W2 - Encoding knowledge in MDP versus learning from data
>
> The trade-off between encoding known problem structure through the MDP and learning it from data is a choice that the designer has to make. Standard NAR implicitly enforces the latter, whereas GNARL enables the designer to encode knowledge through the MDP. This is a new capability that, as we show, yields benefits over learning everything end-to-end. Our view is that, if the basic problem or algorithmic structure is already known, it should be encoded in the MDP. We consider that the role of the neural policy is to make the long-term decisions on which nodes to apply the algorithmic update to, which remains a challenging task as decisions at a particular timestep can have ramifications much later in the episode. We consider that this still qualifies as algorithmic reasoning even though some basic, low-level “primitives” are specified by the designer.
>
> We also faced this choice in the design of each MDP, and we opted for the simplest MDP that could represent the problem, encoding a limited amount of knowledge a priori. Please see Appendices D.4-D.9 (previously D.3-D.8) for explicit details about the transition functions, state features, action masks, and termination rules for each environment. We summarise these below.
>
> For the CO problems, the transition functions are extremely simple, such as adding a node to a set or a tour, and the action masks encode basic validity rules which are part of the problem definition. For the CLRS problems, we similarly use very simple transition functions and action masks, though the choice of transition function can be slightly more involved. An example is in the design of the Bellman-Ford transition function. In the NAR hints, each step represents the state of the algorithm after all edges incident to a particular node have been relaxed. If we allow the MDP transition function to emulate this, then any policy which visits all nodes will succeed in finding the solution. Thus, we instead chose a simpler transition function, being the relaxation of a single edge. This makes the policy non-trivial and thus the problem more interesting from a research perspective. This should be kept in mind when comparing the computational costs of NAR and GNARL on the CLRS-30 problems - we have deliberately limited parallelism to avoid non-trivial policies. In addition, the action masks are defined very simply and permissibly, again so that policies do not become trivial. Omission of the action masks would not preclude the functioning of the framework, but it can help with convergence speed when training with RL. Finally, the rules used for termination are kept as simple and broad as possible. Thus the neural policy still has to solve challenging decision-making problems.
>
> **Changes made:** We have added Appendix D.3 “Encoding Knowledge in the MDP and Role of the Policy” to explain what is being learned by the policy, what is represented by the MDP, and our approach to the design of these elements.

---

> ### Author Response · Authors · 2026-07-21
>
> ### W3 - “Valid by Construction” claim
>
> This concern has been raised by other reviewers, and we appreciate that we must be clearer in our usage of the terms “valid” and “correct”. When we say a solution is “correct”, we mean that it solves the problem - for CLRS a solution is correct if the reference algorithm could have produced it with some node ordering. For CO problems we are attempting to optimise an objective function, so do not use the term “correct”, but have a restriction that the solutions must be “valid”. We use “validity” to mean that the solution does not violate basic problem-defined constraints, such as choosing the same node twice in a tour, or defining a predecessor edge when there is no edge in the graph. For the CLRS problems, we ensure “valid” solutions by restricting predecessors to neighbours in the graph.
>
> Crucially, we do not equate validity and correctness for the CLRS problems, since if action masks were defined by correctness, the policy would become trivial. Hence, while a solution may be valid in the sense that it does not violate the graph structure, it does not necessarily correctly solve the problem. Our claim of “valid by construction” solutions pertains to the former, and does not imply that the solution will be correct. We thank the reviewer for raising this point.
>
> **Changes made:** We have updated Section 4.4 to more clearly define validity and correctness, and also updated several instances throughout the text where the terms were used ambiguously.
>
> ### W3 - CLRS termination rules
>
> Your interpretation is correct, reference-solution information is used for early stopping during inference. In the CLRS-30 benchmark, the number of steps of the algorithm is explicitly given as the number of steps of the reference algorithm. As you state, this does not reflect a deployable procedure in practice, but we adopt an equivalent convention in order to achieve a fair comparison to the existing NAR literature. The GNARL architecture could be easily modified to predict termination through an additional decoder head, but this is out of scope for the paper. We consider that the requirement for an existing solution is not infeasible for the CLRS-30 problems, since these problems are of polynomial complexity, and in fact GNARL can be trained to discover multiple solutions for a single graph while being provided only one single reference solution. More generally, we do not assume we are given a reference solution, and instead rely on the horizon for termination, i.e., reference solutions are not required nor used for the CO problems.
>
> **Changes made:** We have added a new section to Appendix (C.3 “Termination”) to clarify the early termination procedure.
>
> ### W3 - Approximation of graph accuracy in Table 2
>
> Due to lack of details or released code, we were unfortunately not able to reproduce some of the baselines used in the paper, and thus had to rely on an approximation of graph accuracy, which assumes independence of each node prediction. Graph accuracy is a lower bound for solution correctness on BFS and DFS (where multiple solutions are possible), but it is equivalent to solution correctness for Bellman-Ford and MST-Prim, where only one solution is possible (by design of the CLRS-30 Benchmark, not in general). We chose to include these approximations in the table rather than omitting the baselines, as these particular baselines are considered to be SOTA in the NAR literature, and it would not be fair to omit them. In the table we clearly indicate which figures are approximations.
>
> We provide further empirical analysis on the validity of the approximation in Appendix G.1. In this analysis, we demonstrate that the approximated graph accuracy is quite similar to the ground-truth, with the largest difference being 2.4 percentage points, with no consistent trend in over- or under-estimation. Thus we can conclude that the true solution correctness results for Bellman-Ford and MST-Prim would not be competitive with the GNARL performance. For BFS and DFS, the difference between solution correctness and graph accuracy has a larger impact than the node independence assumption, with a maximum difference of 24.2 percentage points. In this case, the addition of these percentage points would not significantly alter the trend of the results in the table, which is that GNARL matches or outperforms all baselines except DNAR.
>
> **Changes made:** We have updated Section 5.1 to clarify the reason for the approximation, and elaborated on the discussion in Appendix G.1 to support our reasoning.

---

> ### Author Response · Authors · 2026-07-21
>
> ### W4 - Instability of reward-driven learning
>
> One of the major advantages of the GNARL framework is that it permits training via either PPO, BC, or any other RL/IL method in principle. The choice of which training approach to use is highly dependent on the problem. For the CLRS problems, there is no naturally defined objective function or reward signal. In this case, BC will tend to outperform PPO, as the hand-designed reward function does not encode the true desired behaviour, but demonstrations are available. In other cases, like the TSP, the problem can be solved with both BC (on sufficiently small problems) and PPO. It is an interesting finding that PPO does not generalise as well as BC in this case. We consider that investigating the underlying reasons for this instability and benchmarking a broad suite of other RL/IL algorithms are important directions for future work, but beyond the scope of this paper.
>
> **Changes made:** We have updated the claims in the abstract and conclusion to highlight the shortcomings of reward-based learning in contrast to fully supervised learning. We have added the full tabulated results for RGC in Appendix D.9 (Table 14).
>
> ### Additional changes
>
> Please see above for our responses to the requested changes that are critical for acceptance. Regarding the non-critical points:
>
> 1. We have thought about additional ablations but could not identify meaningful ones beyond what is already covered in the paper, since the experiments already ablate over: RL or IL as the training method; different reward functions; different MDP designs; using a weak expert to warm-start the RL; and level of stochasticity for the policy. If addressable in the journal timelines, we welcome further suggestions for ablations the reviewer believes would add value.
> 2. The plots for RGC show the uncertainty (95% CI) as the shaded area. We have added the full table in Appendix D.9 (Table 14).
> 3. We have added Section 4.5 to the paper to discuss the computational trade-offs more prominently in addition to the existing discussion in Section 5.3.

---

> ### Comment · Reviewer_tfmH · 2026-07-22
>
> Thank you for the detailed response and substantial revisions. The new material considerably improves the paper’s positioning and transparency. In particular, I appreciate the clearer novelty statement, the explanation of what is encoded in the MDP versus learned by the policy, the distinction between validity and correctness, the numerical RGC results, and the expanded discussion of computational trade-offs.
>
> I consider my concerns about novelty positioning and the previously broad reward-only claims largely addressed. I nevertheless retain one major concern and two smaller qualifications.
>
> ### 1\. Reference-solution early stopping remains a major evaluation concern
>
> The response confirms that Bellman–Ford and MST-Prim use reference-solution information to detect correctness and terminate during test inference. I appreciate that this is now explicit, but I do not consider it equivalent to the CLRS convention of providing a predetermined number of processor steps.
>
> A fixed trajectory length determines when inference ends without inspecting prediction correctness. In contrast, GNARL stops as soon as the current state matches a correct solution. The reported success rate therefore measures whether the policy reaches a correct state before the horizon, rather than whether it returns a correct final state under an executable stopping rule.
>
> This is particularly consequential for Bellman–Ford. In Algorithm 3, selecting an existing edge overwrites $d\_v$ and $\\mathrm{pred}\_v$ without checking whether the edge improves the current distance. A subsequent legal action can therefore corrupt an already correct solution. Oracle stopping may thus affect not only episode length and runtime, but also the reported 93.6% correctness.
>
> **Suggested revision:** Report final-state correctness under a non-oracle fixed horizon or an executable stopping mechanism. Otherwise, relabel the metric as “reaches a correct state before the horizon” and avoid presenting it as deployable final-state correctness.
>
> Please also reconcile the Bellman–Ford horizon definitions: Appendix D.5 gives $P(|V|-1)|E|$, whereas Appendix C.4 states that the horizon is twice the expert trajectory length.
>
> ### 2\. The MDP/policy distinction is clearer, but some claims should remain narrower
>
> Appendix D.3 clearly explains that the policy learns the ordering of node or edge selections, while the transition function implements deterministic update primitives. This addresses my request.
>
> However, the policy does not learn Bellman–Ford relaxation, MST key and predecessor updates, or set/tour construction. It learns when and where to apply these specified operations. The contribution is therefore more precisely described as learning the scheduling or selection of hand-designed algorithmic primitives.
>
> Similarly, claims that GNARL requires “minimal effort,” is straightforward to apply, or abstracts away technical details are not directly demonstrated. Each task still requires problem-specific states, transitions, masks, rewards, horizons, and termination rules.
>
> **Suggested revision:** Describe GNARL as a common framework for learning policies over task-specific update primitives, and present reduced engineering effort as a qualitative design goal rather than an established empirical result.
>
> ### 3\. Validity and baseline comparisons need minor qualification
>
> The revised distinction between validity and correctness is helpful. For tasks such as TSP, “feasible by construction” is appropriate. For CLRS tasks, however, restricting predecessor pointers to graph edges guarantees only local topology consistency, not necessarily a globally valid rooted path or tree.
>
> Appendix G.1 is also useful, but it validates the node-independence approximation only for TripletMPNN, whereas several estimated Table 2 entries concern other architectures. Because node-error dependence may differ across models, this does not establish the accuracy of those estimates. For BFS and DFS, estimated graph accuracy is only a lower bound on solution correctness and cannot establish an exact ranking.
>
> **Suggested revision:** Use “locally topology-consistent by construction” for CLRS tasks, reserve “feasible by construction” for globally enforced constraints, and state that GNARL compares favorably with the reported baseline estimates rather than definitively matching or outperforming every baseline.

---

### Review · Reviewer_5NXr · 2026-07-04

**Summary Of Contributions:**

This paper reframes Neural Algorithmic Reasoning (NAR) as a Markov Decision Process (MDP), enabling the use of MDP tools such as Reinforcement Learning to better learn algorithm trajectories. Instead of imitating the intermediate states of an algorithm as traditional NAR methods do, the paper uses a GNN to predict the action at each step. This naturally supports multiple solutions, ensures valid solutions by construction, and allows the model to go beyond known algorithms by implicitly learning new ones from only a reward signal. Experiments show that the GNARL framework achieves strong results on several CLRS-30 problems, and demonstrates promising performance on NP-hard problems and even on problems for which no expert algorithm exists.

**Strengths**
1. The motivation is clear, and the reframing is both natural and significant. Casting NAR as an MDP broadens the learning paradigm to include both imitation and RL, and enables learning purely from a reward signal, which is valuable in settings where no expert algorithm is available.
2. The formulation and architecture are clear and extensible: action masking prevents invalid solutions, the action space naturally accommodates multiple solutions, and the reward function R = J(s′) − J(s) provides a dense signal that mitigates the credit assignment problem.
3. The empirical results are convincing — competitive on the CLRS-30 problems, effective at recovering multiple solutions, and showing clear potential on NP-hard problems.

**Weaknesses**
1. As noted in the limitations, the MDP relies heavily on knowing the basic steps required to construct a valid solution. For some problems the action space can be kept simple (e.g., selecting a node) with an isolated transition, but this requires design effort and may not generalize as readily as standard NAR, which only requires the intermediate states to be definable. The authors claim that specifying the MDP is no more involved than designing hints for standard NAR, but this claim is asserted rather than substantiated.
2. Table 2 reports only a subset of the CLRS-30 graph problems, leaving the performance on the remaining problems unclear and raising concerns about generalization. The authors also do not analyze why BC alone performs well on these problems — for instance, whether the advantage stems from the action space masks?
3. In "Solving CLRS-30 Problems with RL", designing an effective reward appears difficult, particularly for Bellman-Ford, which further limits the generality of the method.

**Additional Comments:**

N/A.

**Audience:**

Yes

**Audience Explanation:**

The motivation is clear and well posed. The NAR, GNNs, and RL are all of active interest in the community. The problem definition and method are presented clearly, and the MDP reformulation offers a valuable bridge between the NAR and NCO communities, which have largely developed in isolation despite their shared goals. Overall, this is a genuine contribution that would appeal to readers working on algorithmic reasoning and combinatorial optimization.

**Broader Impact Concerns:**

N/A.

**Claims And Evidence:**

Yes

**Claims Explanation:**

The main claims are well supported by the experiments. The authors implement an efficient solution derived directly from the action mask, demonstrate the recovery of multiple solutions in Figure 2, and validate the no-expert setting on the robust graph construction problem. Results on NP-hard problems (MVC, TSP) further show that the proposed method performs comparably to baselines tailored to those specific problems.

My one concern is about generality. The CLRS-30 evaluation covers only four problems, all trained with behavior cloning (BC), whereas reward design under reinforcement learning (RL) is clearly far more difficult. As a result, while the claims are valid, their applicability appears narrower than the framing of the framework suggests.

**Requested Changes:**

1. (critical) Provide a concrete comparison showing that specifying the MDP is no more involved than designing hints, or temper this claim accordingly.
2. (critical) Report results on a broader set of CLRS-30 problems (or justify the current selection), and analyze why BC performs well on these problems while RL does not.
3. Clarify how rewards can be designed for problems beyond path-finding, and state explicitly which problem classes the reward-only setting can and cannot support.

---

> ### Author Response · Authors · 2026-07-21
>
> Thank you for your careful review, please find below our responses to the points that were raised. We hope that this discussion addresses your concerns.
>
> ### Difficulty of designing hints and MDPs
>
> When designing hints in standard NAR, a practitioner must decide which features to include based on the algorithm’s process. At a minimum, this requires any features stored in memory during the execution of the algorithm which cannot be derived from other features. In the MDP design we consider these to be the state features. However, when designing the NAR hints, it is often insufficient (in terms of the resulting accuracy) to use only the minimum set of features, and it is common to include additional derived features in the set of hints in order to attain better results. For example, the DFS algorithm uses [13 different hint/input features](https://github.com/google-deepmind/clrs/blob/master/clrs/_src/specs.py#L275-L289), of which GNARL uses only 3 (with an additional 2 default features for the phase tracking). While the design of the MDP transition function requires design effort, so too does the selection of extra hints for use in NAR. We accept that it is difficult to provide quantitative evidence and to assess the design effort beyond practitioner anecdotes.
>
> **Changes made:** We have softened the claim to state that, like NAR, GNARL also requires encoding practitioner expertise.
>
> ### Choice of CLRS problems
>
> The scope of our evaluation is restricted to graph problems (hence the G in GNARL). Among the graph algorithms in CLRS-30, we have chosen a subset with an increasing range of difficulty as quantified by SOTA models’ performance. More specifically, BFS proves to be consistently very easy, DFS has highly mixed outcomes, and both Bellman-Ford and MST-Prim are solved with relatively high *node* accuracy but poor *graph* accuracy.
>
> There is a precedent in the literature to focus on a subset of graph algorithms [1,2,3]. In particular, given  the 100% success rate of the DNAR approach [2], one may consider the CLRS-30 problems to be solved, and that research resources are better spent on exploring the generalisation of the NAR ethos beyond P-hard problems. GNARL can certainly be extended to other graph algorithms and other CLRS-30 algorithms more broadly. However, we leave this for future work, as our notable achievements relative to the state-of-the-art in NAR are on NP-hard problems.
>
> **Changes made:** We have clarified that the evaluation is restricted to graph problems and elaborated on the selection of the CLRS-30 problems.
>
> ### Better performance of BC relative to RL on CLRS
>
> In our view, it is unsurprising that BC performance is superior to RL for the CLRS-30 problems, as the CLRS-30 problems do not have specified objective functions. In order to apply an RL method to a problem, we require a reward function to maximise, which in GNARL is derived from the objective function. We include Section 5.1.2 to illustrate that, while it is possible to attempt to specify a reward function, it is unlikely that such a hand-designed reward function will outperform BC, which is directly minimising the distance between the policy’s and the expert’s distribution. Designing an appropriate reward function is a well-known challenge in the RL literature [4].
>
> We consider that the training method should be chosen based on the properties of the problem (i.e., presence of an expert, knowledge of an objective function). It may be possible to learn a reward function from the expert algorithm using inverse RL [5], but this is also beyond the scope of the paper. The discrepancy in performance is a property of the methods, which are optimising for different criteria, and we argue that GNARL’s ability to be trained with either of these methods, depending on which is most appropriate for the problem at hand, makes GNARL *more* general, rather than less.
>
> **Changes made:** We have extended the discussion in Section 5.1.2 to cover these points.
>
> ---
>
> [1] Minder, J., Grötschla, F., Mathys, J., & Wattenhofer, R. (2023). SALSA-CLRS: A sparse and scalable benchmark for algorithmic reasoning. In Learning on Graphs Conference.
>
> [2] Rodionov, G. and Prokhorenkova, L., 2025. Discrete Neural Algorithmic Reasoning. In International Conference on Machine Learning.
>
> [3] Kujawa, Z., Poole, J., Georgiev, D., Numeroso, D., Fleischmann, H., and Liò, P. (2025). Neural algorithmic reasoning with multiple correct solutions. In KDD Workshop on Machine Learning on Graphs in the Era of Generative Artificial Intelligence.
>
> [4] Booth, S., Knox, W.B., Shah, J., Niekum, S., Stone, P. and Allievi, A. (2023). The perils of trial-and-error reward design: misdesign through overfitting and invalid task specifications. In Proceedings of the AAAI Conference on Artificial Intelligence.
>
> [5] Ng, A. Y., & Russell, S. (2000). Algorithms for inverse reinforcement learning. In International Conference on Machine Learning.

---

### Review · Reviewer_J59D · 2026-07-12

**Summary Of Contributions:**

This paper proposes GNARL, a framework that reformulates neural algorithmic reasoning as sequential decision-making in a Markov decision process. Instead of predicting intermediate algorithmic states through the conventional encode-process-decode NAR pipeline, GNARL uses an encode-process-act architecture in which a graph neural network produces a policy over node-selection actions. The formulation supports both imitation learning from expert trajectories and reinforcement learning from reward signals.

The paper evaluates the framework on polynomial-time graph algorithms from CLRS-30, including BFS, DFS, Bellman-Ford, and MST-Prim, as well as NP-hard or combinatorial optimization problems including minimum vertex cover and TSP. It also considers robust graph construction, where no strong expert algorithm is assumed to be available. The experiments suggest that the proposed construction can generate feasible terminal solutions without an additional repair stage, represent multiple valid solutions, outperform several previous NAR baselines on the considered combinatorial problems, and remain applicable when only a reward signal is available.

The main strengths are:

**1** The connection between NAR trajectories and MDP formulations is conceptually useful and provides a clear bridge between the NAR and graph reinforcement-learning literatures.

**2** The framework is evaluated on a relatively broad collection of problems rather than a single highly specialized domain.
**3** The paper evaluates both behavioral cloning and PPO, including settings with exact experts, weak experts, and no expert.
**4** The emphasis on whole-solution correctness rather than only node-wise accuracy is appropriate.
**5** The paper is generally clear about the distinction between a general NAR-oriented framework and specialized neural combinatorial optimization solvers.


The main weaknesses are:

**1**  Several equations and pieces of pseudocode appear inconsistent with either the stated method or standard implementations. These issues seem correctable, but currently make parts of the method difficult to reproduce exactly.

**2**  Some claims concerning “validity by construction” and comparisons to existing methods should be stated more precisely.

**3** Some baseline comparisons rely on estimated quantities or results taken from different evaluation pipelines, limiting the strength of direct conclusions.

**4** The framework requires a manually designed transition system and often substantially longer trajectories than standard NAR, which limits the practical generality suggested by some of the presentation.

Overall, I find the central contribution valuable and the empirical evidence promising. I recommend a **minor revision**, primarily to correct technical inconsistencies and clarify the scope of the claims.

**Audience:**

Yes

**Audience Explanation:**

The paper should be of interest to researchers working on neural algorithmic reasoning, graph neural networks, imitation learning, reinforcement learning, and learning for combinatorial optimization.

The work addresses a meaningful limitation of conventional NAR: supervised imitation of one predefined sequence of intermediate states can be awkward when multiple solutions are correct, when feasibility must be maintained globally, or when no sufficiently strong expert trajectory is available. Expressing the same problems as MDPs is a natural but useful reframing, and the paper explains the resulting design choices in a manner that could inform future work beyond the specific architecture evaluated here.

The comparison between behavioral cloning and PPO is also informative. In particular, the finding that behavioral cloning often provides substantially better size generalization, while PPO permits learning without expert trajectories, is relevant to the design of future neural algorithmic systems. The substantial increase in sequential trajectory length relative to standard NAR is likewise an important negative result and helps characterize the trade-off introduced by the proposed formulation.

Although many individual components—GNN policies, constructive MDPs, action masking, behavioral cloning, and PPO—are established, their integration as a general NAR reformulation and the breadth of the evaluation make the work relevant to at least a meaningful subset of the TMLR readership.

**Broader Impact Concerns:**

I do not see a broader-impact concern that would prevent publication or require an extensive dedicated analysis. The work studies general algorithmic and combinatorial optimization methods and does not directly involve sensitive personal data, human-subject experiments, surveillance, or high-risk automated decision-making.

A short broader-impact statement would nevertheless be useful. It could mention that learned combinatorial optimization policies may be deployed in areas such as infrastructure, logistics, or network design, where approximate or distribution-shifted outputs can have real operational consequences. Although the construction may enforce some feasibility constraints, it does not guarantee optimality or robust performance outside the evaluated distributions. The authors could therefore caution against interpreting feasibility by construction as a general safety or reliability guarantee.

The statement could also briefly acknowledge computational cost. Some experiments require long sequential trajectories and substantial training time, so future applications should compare the energy and resource cost against conventional algorithms and heuristics.

**Claims And Evidence:**

Yes

**Claims Explanation:**

The paper provides reasonably broad empirical support for its principal claims. It reports experiments on four CLRS-30 problems, two established combinatorial optimization problems, and robust graph construction. The evaluation includes out-of-distribution graph sizes, multiple random seeds, comparisons against conventional NAR approaches, and analyses of solution correctness, trajectory length, training time, and inference time. The distinction between behavioral cloning and reward-only reinforcement learning is also experimentally investigated.

The evidence is particularly convincing for the following narrower conclusions:

Reformulating an algorithm as a sequential action-selection problem can ensure that certain elementary feasibility constraints are respected by the terminal construction.
The resulting framework can represent different valid action trajectories rather than being tied to one canonical node ordering.
On the specific MVC and TSP datasets considered, GNARL substantially improves over the cited NAR baselines.
Reward-only learning is possible in the proposed framework, although its generalization is often weaker than behavioral cloning.
The approach can be instantiated for a problem without a strong expert, as illustrated by robust graph construction.

However, several qualifications are needed.

First, “valid by construction” does not always mean “algorithmically correct by construction.” Action masking prevents operations such as revisiting a TSP node or selecting a nonexistent edge, but correctness of BFS, DFS, shortest-path, or minimum-spanning-tree outputs still depends on the learned policy and the termination criterion. The paper generally acknowledges that feasibility does not imply optimality, but the wording should distinguish structural feasibility from full solution correctness more consistently.

Second, the comparison with some CLRS baselines uses graph accuracy estimated as node accuracy raised to the number of nodes. This calculation implicitly treats node errors as independent, which will generally not hold. It may serve as a rough heuristic or lower-bound-style illustration in selected cases, but it should not be presented as a directly comparable measured result.

Third, some experiments compare against values reported in prior work rather than all methods being retrained under a unified implementation and model-selection protocol. This does not invalidate the findings, but claims of statistical superiority should be limited to genuinely matched comparisons.

Finally, several apparent equation and pseudocode errors need correction before the evidence is fully reproducible. These include the direction of the KL divergence used for behavioral cloning, the BFS expert policy in Algorithm 9, the RGC edge-budget expression, and smaller notation inconsistencies. I regard these primarily as presentation or specification errors rather than evidence that the reported experiments are invalid, but the authors should explicitly confirm the behavior of the released implementation.

Subject to these corrections and clarifications, the main claims are adequately supported.

**Requested Changes:**

**1.** Correct or clarify the behavioral-cloning KL objective.

Equation (4) minimizes

$D_{KL}
\left(
\pi(\cdot \mid s)
\,\|\,
\pi_{\mathrm{expert}}(\cdot \mid s)
\right)$.

However, the expert distributions described in the paper assign zero probability to many actions. Consequently, this reverse-KL objective becomes infinite whenever the learned policy assigns nonzero probability outside the support of the expert policy. Standard distributional behavioral cloning would more commonly minimize

$D_{\mathrm{KL}}
\left(
\pi_{\mathrm{expert}}(\cdot \mid s)
\,\|\,
\pi(\cdot \mid s)
\right)$,

which is equivalent, up to the entropy of the expert policy, to minimizing cross-entropy with the expert distribution.

Please state the objective actually used in the implementation, correct the equation if necessary, and explain whether any smoothing or masking was applied to actions with zero expert probability.

**2.**  Resolve the apparent error in the BFS expert policy.

In Algorithm 9, PhaseTwoPolicy appears to define the eligible neighbors as nodes satisfying $\mathrm{reach}_j = 1$.

Given the state transition and the interpretation of reach, this appears to select already reached nodes, whereas BFS expansion would normally select unreached neighbors. Algorithm 10 for DFS uses the condition $\mathrm{reach}_j = 0$, which further suggests that the BFS condition may be a typographical error.

**3.**  Please correct the pseudocode or explain the state semantics if the current condition is intentional. Since this expert policy is used to generate BFS demonstrations, the authors should also confirm that the released implementation uses the intended condition.

Correct the robust graph construction budget formula.

Appendix D.8 defines the edge-addition budget as

$\ell=\left\lceil
\frac{2\tau}{|V|^2 - |V|}
\right\rceil$.

For $\tau = 0.05$, this expression appears to yield a budget of one edge for essentially all graph sizes considered, which does not seem consistent with interpreting τ as the fraction of possible undirected edges to add. The intended expression may instead involve multiplication by the number of possible edges.

The decrement of the remaining-budget feature in Algorithm 7 should also be checked for the corresponding factor of two. Please correct the formulas and explicitly report the edge budget used for each evaluated graph size.

---

> ### Author Response · Authors · 2026-07-21
>
> Thank you for your very detailed scrutiny of the paper. In particular, we appreciate your precision in picking up several minor errors, which we have corrected in our revision. Following checks, we concluded that these are presentational errors in the paper, but that the correctness of the implementation in our source code is unaffected. Therefore, these presentational errors thankfully do not have an impact on the results of the paper.
>
> ### Validity by construction claim
>
> We agree that this claim was not clearly scoped, and we have revised the paper to more precisely define what we mean by this. This was also discussed in our response to Reviewer `tfmH` (W3 - “Valid by Construction” claim), which we reproduce below:
>
> > This concern has been raised by other reviewers, and we appreciate that we must be clearer in our usage of the terms “valid” and “correct”. When we say a solution is “correct”, we mean that it solves the problem - for CLRS a solution is correct if the reference algorithm could have produced it with some node ordering. For CO problems we are attempting to optimise an objective function, so do not use the term “correct”, but have a restriction that the solutions must be “valid”. We use “validity” to mean that the solution does not violate basic problem-defined constraints, such as choosing the same node twice in a tour, or defining a predecessor edge when there is no edge in the graph. For the CLRS problems, we ensure “valid” solutions by restricting predecessors to neighbours in the graph.
> >
> > Crucially, we do not equate validity and correctness for the CLRS problems, since if action masks were defined by correctness, the policy would become trivial. Hence, while a solution may be valid in the sense that it does not violate the graph structure, it does not necessarily correctly solve the problem. Our claim of “valid by construction” solutions pertains to the former, and does not imply that the solution will be correct. We thank the reviewer for raising this point.
> >
> > **Changes made:** We have updated Section 4.4 to more clearly define validity and correctness, and also updated several instances throughout the text where the terms were used ambiguously.
>
> ### Graph accuracy figure
> While we were unable to calculate a solution correctness figure for some baselines due to lack of available code, we have used an approximation of graph accuracy based on the reported node accuracy. For Bellman-Ford and MST-Prim the graph accuracy is equivalent to solution correctness as there is only one possible solution, and the approximation of graph accuracy is reasonable based on empirical analysis in Appendix G.1. For BFS and DFS, the solution correctness can be higher than the graph accuracy of a single solution, but the under-approximation does not change the trend of the results. This was also discussed in our response to Reviewer `tfmH` (W3 - Approximation of graph accuracy in Table 2), which we reproduce below:
>
> > Due to lack of details or released code, we were unfortunately not able to reproduce some of the baselines used in the paper, and thus had to rely on an approximation of graph accuracy, which assumes independence of each node prediction. Graph accuracy is a lower bound for solution correctness on BFS and DFS (where multiple solutions are possible), but it is equivalent to solution correctness for Bellman-Ford and MST-Prim, where only one solution is possible (by design of the CLRS-30 Benchmark, not in general). We chose to include these approximations in the table rather than omitting the baselines, as these particular baselines are considered to be SOTA in the NAR literature, and it would not be fair to omit them. In the table we clearly indicate which figures are approximations.
> >
> > We provide further empirical analysis on the validity of the approximation in Appendix G.1. In this analysis, we demonstrate that the approximated graph accuracy is quite similar to the ground-truth, with the largest difference being 2.4 percentage points, with no consistent trend in over- or under-estimation. Thus we can conclude that the true solution correctness results for Bellman-Ford and MST-Prim would not be competitive with the GNARL performance. For BFS and DFS, the difference between solution correctness and graph accuracy has a larger impact than the node independence assumption, with a maximum difference of 24.2 percentage points. In this case, the addition of these percentage points would not significantly alter the trend of the results in the table, which is that GNARL matches or outperforms all baselines except DNAR.
> >
> > **Changes made:** We have updated Section 5.1 to clarify the reason for the approximation, and elaborated on the discussion in Appendix G.1 to support our reasoning.

---

> ### Author Response · Authors · 2026-07-21
>
> ### Requested changes
>
> Regarding the requested changes, thank you very much for finding these errors. As mentioned above, these errors were presentational, and did not affect the results of the paper.
>
> **Changes made:** We have corrected these as shown below:
> 1. We can confirm that our implementation minimises $D_{\mathrm{KL}} \left(\pi_{\mathrm{expert}}(\cdot \mid s),|,\pi(\cdot \mid s)\right)$. No smoothing or masking was applied to the expert actions. Section 3 was changed accordingly.
> 2. You are correct, line 14 of Algorithm 9 should have the condition $\texttt{reach}_j = 0$.
> 3. Apologies, the edge budget should be defined as $\ell =\lceil \tau (|V|^2 - |V|)/2 \rceil$, and there is indeed a missing factor of 2 in Algorithm 7. We have listed the edge addition budget for each graph size in the tabulated RGC results in Table 14.
>
> ### Broader impact statement
>
> **Changes made:** We have added a broader impact statement, as we agree that it is useful to discuss the potential implications of this work in contexts where distribution shift has strong consequences.